# MEDLINE search retrieval issues: A longitudinal query analysis of five vendor platforms

**C. Sean Burns** [1]*, **Tyler Nix** [2], **Robert M. Shapiro, II** [3], **Jeffrey T. Huber** [1]

1 School of Information Science, University of Kentucky, Lexington, Kentucky, United States of America,
2 Taubman Health Sciences Library, University of Michigan, Ann Arbor, Michigan, United States of America,
3 Robert M. Fales Health Sciences Library - SEAHEC Medical Library, South East Area Health Education Center, Wilmington, North Carolina, United States of America

* sean.burns@uky.edu

**Data Availability Statement:** The data underlying the results presented in the study are available from https://github.com/cseanburns/medline-longitudinal.

## Abstract

This study compared the results of data collected from a longitudinal query analysis of the MEDLINE database hosted on multiple platforms that include PubMed, EBSCOHost, Ovid, ProQuest, and Web of Science. The goal was to identify variations among the search results on the platforms after controlling for search query syntax. We devised twenty-nine cases of search queries comprised of five semantically equivalent queries per case to search against the five MEDLINE database platforms. We ran our queries monthly for a year and collected search result count data to observe changes. We found that search results varied considerably depending on MEDLINE platform. Reasons for variations were due to trends in scholarly publication such as publishing individual papers online first versus complete issues. Some other reasons were metadata differences in bibliographic records; differences in the levels of specificity of search fields provided by the platforms and large fluctuations in monthly search results based on the same query. Database integrity and currency issues were observed as each platform updated its MEDLINE data throughout the year. Specific biomedical bibliographic databases are used to inform clinical decision-making, create systematic reviews, and construct knowledge bases for clinical decision support systems. They serve as essential information retrieval and discovery tools to help identify and collect research data and are used in a broad range of fields and as the basis of multiple research designs. This study should help clinicians, researchers, librarians, informationists, and others understand how these platforms differ and inform future work in their standardization.

## Introduction

Bibliographic databases are used to identify and collect research papers and function as a critical part of scientific investigation [1]. Studies that employ bibliographic databases include research on information literacy, bibliometrics/scientometrics, information seeking, systematic reviews, and meta-analyses [2]. In particular, PubMed and MEDLINE are used to inform

**Funding:** This project was supported by a 2018 Summer Faculty Research Fellowship from the College of Communication and Information, University of Kentucky. The authors received support in the form of one year of access to MEDLINE on Ovid from Wolters Kluwer (https://www.ovid.com/). The funders had no role in study design, data collection and analysis, decision to publish, or preparation of the manuscript.

**Competing interests:** Wolters Kluwer provided a one-year subscription to OVID MEDLINE. This was used to collect the COVID-19 data but not the longitudinal data. This does not alter our adherence to PLOS ONE policies on sharing data and materials. Wolters Kluwer or its representatives have not been involved in this study or offered any comments on it.

clinical decision-making in the health professions [3] and to construct knowledge bases for clinical decision support systems [4].

Research on search queries that inform bibliographic database development or on how queries influence information retrieval results were once common lines of inquiry, [5] but these studies have subsided in recent decades [6]. Search query research has largely shifted away from a Boolean model of information retrieval and has focused on ranked-based keyword systems [7] or on database coverage [8–11].

Researchers, librarians, information scientists, and others rely on bibliographic databases to conduct research, instruct future information and other professionals on how to conduct literature searches, and assist those with information needs to locate and access literature [12–15]. Furthermore, these databases have standard rules to describe research papers, and are structured using controlled vocabularies, in order to make searching for information more precise and comprehensive [16, 17].

Fine control over bibliographic searching and documentation of search strategies, which are reported in systematic reviews, allow for the replication and reproduction of searches. In the broader scientific community, the replication and reproduction of research, or lack thereof, has garnered increased attention recently [18, 19]. Additional scrutiny has been given to the replication of prior studies [20]. This is true for systematic reviews and other research that rely on citation or bibliographic records, but in this domain, the evaluation of scientific rigor is centered around the reproducibility of search strategies. The Preferred Reporting Items for Systematic Reviews and Meta-Analyses (PRISMA) Guidelines [21] and the Cochrane Handbook for Systematic Reviews of Interventions provide examples for scholars in their reporting of methods and organizing of reviews [22].

Unlike general search engines, bibliographic databases, such as those available on EBSCO-host, ProQuest, Web of Science, Scopus, Ovid platforms, and others are designed to use structured bibliographic records instead of full-text sources to create search indexes. These bibliographic records contain fields for providing discovery and access, such as author name fields, document title fields, publication title fields, and date of publication fields [16]. In certain specialized databases, these records are often supported by a set of thesaurus terms, or a controlled vocabulary, such as the Medical Subject Headings (MeSH). Their goal is to describe the subject matter of works or records that are indexed in the bibliographic database to assist in consistent and predictable information retrieval.

Controlled vocabularies and thesauri are meant to provide users with a high level of control over the search and retrieval process [23, 24] and may be available across multiple platforms or interfaces from different vendors [25]. The MeSH thesaurus is freely available on the U.S. National Library of Medicine's (NLM) PubMed website, and is used to search MEDLINE on other platforms such as EBSCOhost, Ovid, ProQuest, and Web of Science. These commercial vendors provide access to bibliographic data from MEDLINE, and the corresponding MeSH thesaurus, and add features on their respective platforms beyond what NLM has already provided. The added features are based upon the providers' unique user interface, search features, ability to link to library collections via proxies, related additional database content, or searching multiple databases on a specific platform in a single search session.

However, these added features may create some differences in searching and in the search results [25, 26]. For example, MEDLINE can be searched using PubMed, which is defined by the nearly 6000 publication titles it indexes, structure of its bibliographic records, use of MeSH in those records, and its freely available search interface on the web. When a vendor provides access to MEDLINE, they start with the MEDLINE system but create a customized subscription-based version that includes a different interface, slightly different search fields, search operators, and other features.

The differences among platforms have been recognized as important in the systematic review literature in the biomedical sciences, and the forthcoming PRISMA-S Search Reporting Extension recommends that systematic reviewers report which platform, interface, or vendor is used for each database searched [27]. However, the implications in how bibliographic records are queried on different platforms are not well understood [28, 29]. For example, even though PubMed/MEDLINE, ProQuest/MEDLINE, EBSCOhost/MEDLINE, Ovid/MEDLINE, and Web of Science/MEDLINE are built on the same MEDLINE database, it is not fully known how the features that are added by these vendors impact search and retrieval on their respective platforms. Even when there is some transparency, such as with PubMed [30], these systems are complicated and differences with other systems are not well understood.

Although the choice of platforms impacts potential source coverage, it is not known how searching a single database like MEDLINE but on different platforms might affect source coverage. That is, searchers use different content-based databases to conduct literature searches [31–33] to avoid missing relevant studies [9, 34], but there is no research to show whether it is necessary to search multiple MEDLINE-based databases to prevent missing relevant studies. If it is important to access multiple MEDLINE-based databases to expand source coverage, then this is important in cases where data from past research is collected, analyzed, and synthesized based on published and/or gray literature, such as in systematic reviews or meta-analyses [10, 35].

In addition to source coverage, it is important to provide a detailed description of the search strategies that are used to search databases so that others may investigate the quality of a search strategy or replicate it in the future [36]. However, research that supports this only refers to MEDLINE as a source and not to MEDLINE from any specific vendor. Not distinguishing which MEDLINE platform is searched assumes consistency between MEDLINE platforms: for example, that using MEDLINE on PubMed is equivalent to using MEDLINE on Ovid, EBSCOhost, ProQuest, or Web of Science. This may have ramifications for those researchers leading clinical trials or conducting bench-side research, and who have to rely on published literature and conduct intensive literature searches when systematic reviews on their topic are not available.

Even if search sessions are methodical and well documented, vendor systems often operate as black boxes (i.e., the technology is not well documented) and it becomes only possible to infer how different systems operate by comparing multiple implementations [37]. Little is known about what actual principles are applied by platform vendors in indexing bibliographic records or what specific types of algorithms are used to rank results when sorted by system-defined relevance. This is a commonly known problem among search engines, but it is problematic in bibliographic resources purchased by libraries [38, 39].

Interface, indexing, and retrieval differences also impact reproducibility and replication, which are important aspects of the scientific process, evidence-based medicine, and the creation of systematic reviews [32, 40–43]. Although the NLM maintains the MEDLINE records and provides free (federally subsidized) access to them through the PubMed website, they also license these records to vendors to host on their own platforms. Furthermore, although these systems operate from the same MEDLINE data file, database vendors apply their own indexing technologies and their own search interfaces, and it is possible that these alterations influence different search behaviors and retrieval results sets [44, 45]. This may be problematic if platform differences are not commonly understood, communicated in vendor reports, or among research team members using them, and if the separate platforms are unable to replicate results based on the same data files that are used across them.

There is little scientific evidence that shows any discrepancies between MEDLINE platforms. Some studies have tested reproducibility across systems but not across MEDLINE-

based systems [34]. Instead, many studies compare some aspect of the database collection or the recall and precision on the retrieval results sets in these systems [46–48]. However, the focus is not often on the query syntax used even though this has been highlighted as an important problem. One study investigated variations among different interfaces to the Cumulative Index for Nursing and Allied Health Literature (CINAHL) database, and reported reproducible search strategies except for queries that contained subject-keyword terms [28]. In our prior paper [49] we found that queries searched in MEDLINE across different platforms resulted in search result discrepancies after controlling for the search query. This paper builds on that work and examines differences over time by evaluating longitudinal data, which is a critical factor in replicating bibliographic database search results. Here we ask the following research questions:

1. How do search results among MEDLINE-based bibliographic database platforms vary over time after controlling for search query syntax?

2. What explains the variance among search results among MEDLINE-based bibliographic database platforms after controlling for search query syntax?

To answer these questions, our analytical framework is based on the concepts of *methods* and *results reproducibility* [50]. Methods reproducibility is "the ability to implement, as exactly as possible, the experimental and computational procedures, with the same data and tools, to obtain the same results" and results reproducibility is "the production of corroborating results in a new study, having followed the same experimental methods (A New Lexicon for Research Reproducibility section, para. 2). We do not apply the concept of inferential reproducibility in this paper since this pertains to the conclusions that a study makes based on the reproduced methods, and this would largely be applicable if we investigated the relevance of the results based on an information need rather than, as we do, focus solely on the reproducible results of search queries and the records produced by executing those queries.

## Materials and methods

We designed 29 cases of search strategies. Each case comprised five (5) semantically similar or equivalent queries to perform searches in five MEDLINE-based platforms for a total of 145 searches. We collected longitudinal data (October 2018—September 2019) for each case after two pilot runs in August and September 2018. The five platforms included what is now *legacy* PubMed/MEDLINE (PMML), which has undergone an interface update that applies new search algorithms [51], ProQuest/MEDLINE (PQML), EBSCOhost/MEDLINE (EHML), Web of Science/MEDLINE (WSML), and Ovid/MEDLINE (OML). The data is based on search result counts for each search strategy in the 29 cases and were collected at the mid-point of each month. Fig 1 illustrates the general process we used to collect data over the twelve-month period, and Fig 2 highlights the search parameters and values used in the searches.

The search queries in each case, tested in the pilot studies, were designed to be semantically and logically equivalent to each other per case. Throughout the paper we refer to the semantically equivalent queries as cases. Differences between queries within cases were made only to adhere to the query language and syntax required by each MEDLINE platform. Table 1 provides an example search for case #10 for October 2018. Each of the queries in this case were designed to search each of the five MEDLINE platforms for the MeSH term *dementia*, chosen, like the other terms, for simplicity and for its representation in MEDLINE, which has two MeSH branches (topical and analytical structure of the MeSH tree [52]), to explode the term,

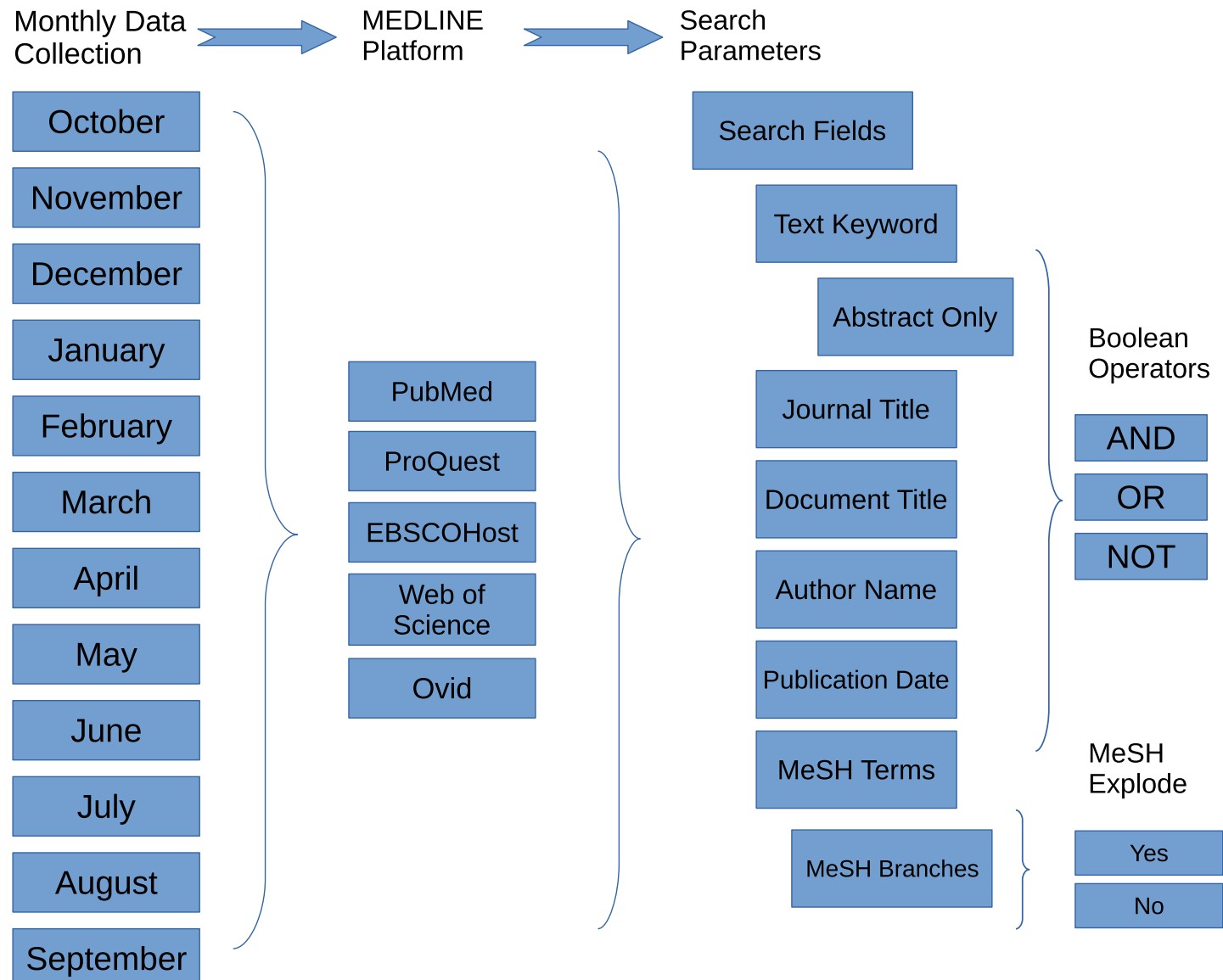

**Fig 1. Data collection method.** Diagram outlining the research process used to collect data. Each month we ran 145 searches (29 cases) on the five MEDLINE platforms using the specified search parameters.

and to limit results by publication date from 1950 to 2015. The last column reports the number of records that were retrieved for each of the platforms for the month and year that data were collected.

We designed our 29 cases using basic search fields to examine search retrieval counts on each of the five MEDLINE platforms; therefore, our queries were designed to be short and logically clear to enhance the comparability of the results. For example, Table 1 describes a simple MeSH field search, limited by publication date range. Despite this simplified search, our prior findings indicated that a wide range of records were retrieved by the different platforms, even when MeSH terms were added to the strategies/queries. On this basis, we created 29 cases then

## Search Parameters

## Search Values

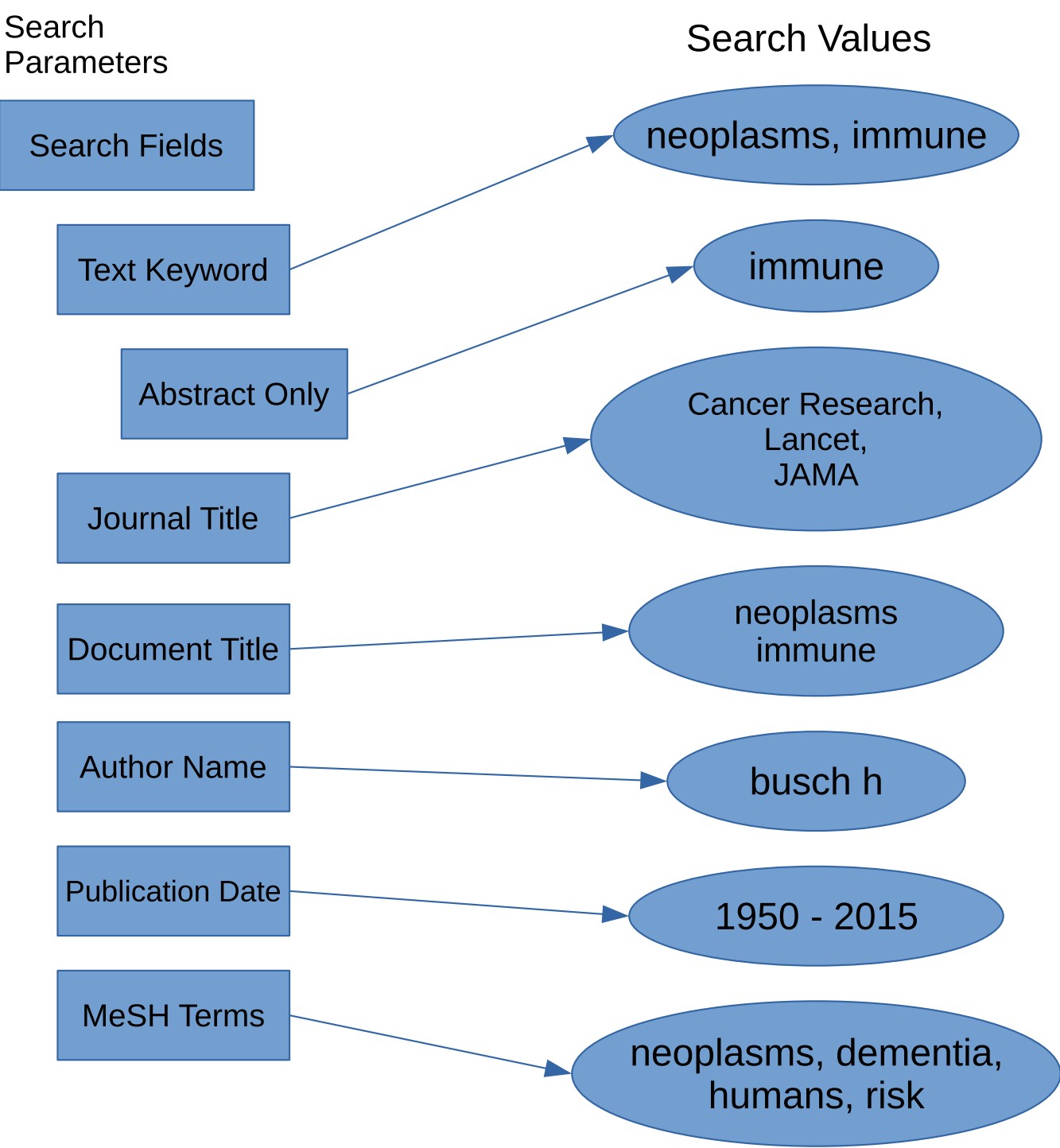

**Fig 2. Search parameters.** Comprehensive list of search values for the search parameters used for data collection.

to test a broad range of basic search strategies. We stress that the queries were not designed to test user relevance or user needs, which may range from simple searches to complex, multi-part queries designed for systematic review research. Rather, our queries were designed to contain some part or combination of the following search fields:

**Table 1. Case #10 showing semantically equivalent but syntactically different queries for five MEDLINE platforms.**

| Platform | Case #10 | 10-2018 |
|---|---|---|
| PMML | `"dementia"[MH] AND 1950:2015[DP]` | 134217 |
| PQML | `MESH.EXACT.EXPLODE ("dementia") AND YR(1950-2015)` | 132593 |
| EHML | `MH("dementia+") AND YR 1950-2015` | 132599 |
| WSML | `MH:exp = ("dementia") AND PY = (1950-2015)` | 132590 |
| OML | `1. EXP dementia/ 2. limit 1 to YR = 1950-2015` | 132593 |

- Keywords

- Specific fields

- MeSH terms with one (branch) tree number

- MeSH terms with more than one (branch) tree number

- MeSH terms that were exploded

The values that we chose for our search strategies were based on the following criteria: search terms must be well represented in MEDLINE in order to retrieve enough records to examine how results might vary across MEDLINE platforms; the specific MeSH terms must include at least one term with only one branch in the MeSH thesaurus and one MeSH term with more than one branch in order to examine differences when the terms are exploded. We limited MeSH terms to four that included two primary terms, *neoplasms* and *dementia*, that were combined with secondary terms in some cases. The secondary MeSH terms included *humans* and *risk* and were searched with the primary MeSH terms with Boolean operators (AND, NOT) in order to compare search count results to queries constructed specifically for each platform that included Boolean processing. The journal titles we used in our queries were chosen to ensure that results would include records from the titles based on the topics we searched, and therefore we chose well established titles in the biomedical sciences that fit our query topics and these include *Cancer Research*, *JAMA*, and *Lancet*. For those cases that include publication dates, we chose a broad publication date range to retrieve a substantial number of results, and we ended the publication date range to records with publication dates up to and including 2015 so that we could limit the effect of records that have been newly added to the platforms. Specifically, nine of our cases included publication date ranges, as demonstrated in Table 1. We added date ranges to control differences among the platforms. Most queries also included at least one Boolean operator. Figs 1 and 2 show that for each month during data collection, we searched the five MEDLINE platforms using a range of search parameters. For a more detailed examination, all search queries and search result counts are included in S1 Data.

To answer our first research question, our analysis is based on a comparison of search result counts in each case and on modified z-scores ($m_i$) based on the range of counts for each platform within each case for the time period. The modified z-score is a version of the standard z-score, but is more robust against outliers [53]. We used the modified z-score to locate deviations around results from the PubMed MEDLINE platform instead of the distribution center (mean or median) in order to highlight deviations from PubMed. We use PubMed as the reference platform since this platform and the MeSH vocabulary are provided by the National Library of Medicine. Otherwise, this is an arbitrary reference point and we could have chosen any of the other platforms as reference points since they all presumably are based on the same underlying MEDLINE data. We also define our search result count outliers as any modified

z-score that deviates more than ±3.5 from PubMed, as recommended by Iglewicz and Hoaglin [53]. However, statistical outliers and practical or clinical outliers are different issues. The modified z-score will only highlight the former, but any difference in counts will help answer how MEDLINE platforms deviate from each other. To answer our second research question, we inspected several records from our cases in order make initial inferences about reproducibility issues that might be a result of how the databases index bibliographic records or respond to the queries that we designed.

## Results and discussion

The data gathered in the majority of our searches (20 cases of five searches each for a total of 100 queries) did not include limits on publication dates. However, we did include date delimited searches for nine cases of five searches each for a total of 45 queries. We used publication date limits from 1950–2015 (see Table 1 for an example case) for these search queries. Thirty-nine of the publication date restricted queries returned different search results from October 2018 to September 2019, indicating potential changes either in the bibliographic records nearly five years after the last publication or potential changes with the search mechanisms used among the MEDLINE platforms. This discrepancy yielded insights for the differences we found across the retrieved records. For queries that included a limit on the publication date, our findings break the notion that the bibliographic universe (the set of all pertinent published works) is stable years after works have been published. First, based on a broad overview (Fig 3),

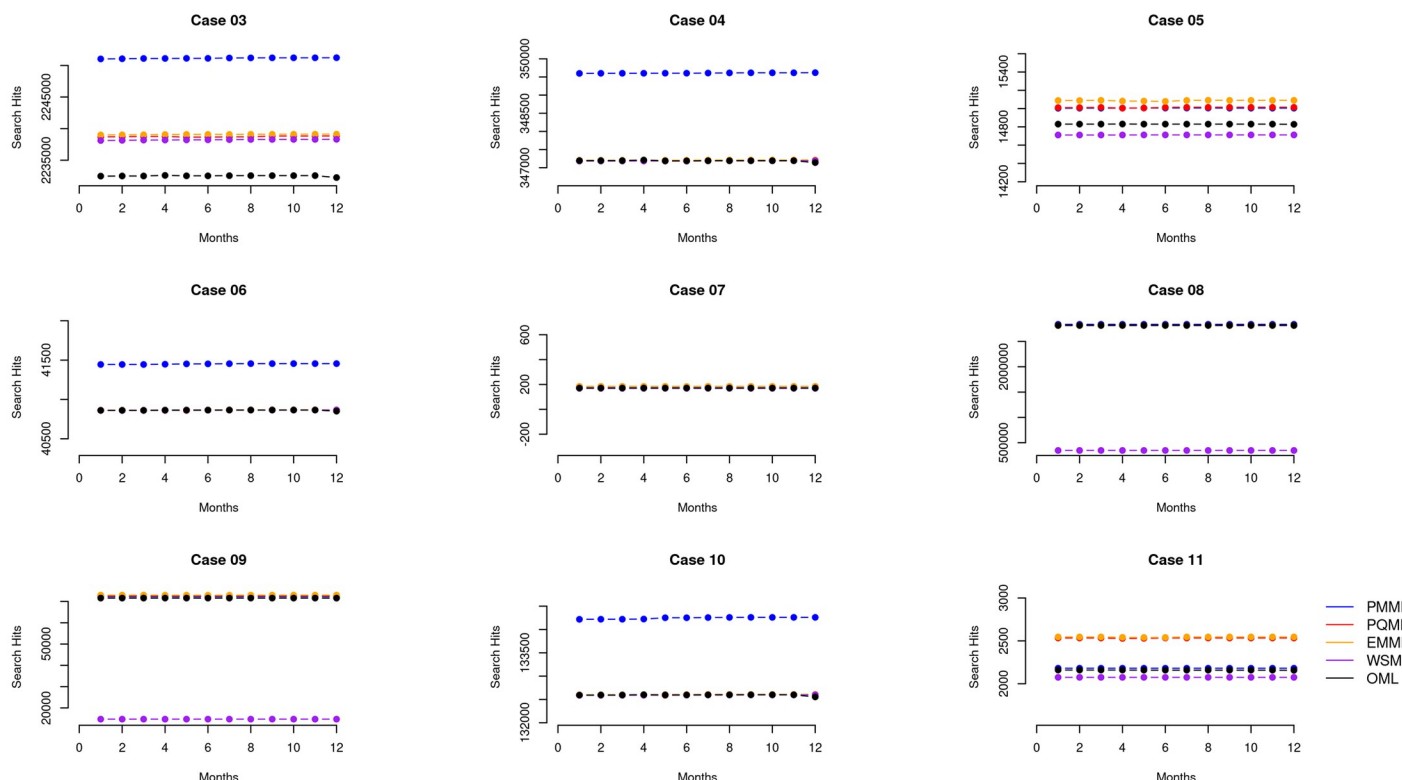

**Fig 3. Macro view of cases #03–#11.** A macro perspective of search result counts for cases #03-#11 that were restricted by publication date. The individual plots are case by case and each plot compares the count of retrieved items for the five MEDLINE platforms across the twelve months of data collection. The macro perspective highlights differences among the ranges of records retrieved among the platforms in each case but it does not capture the detailed variation or trends that occurs when comparing retrieved results among the platforms.

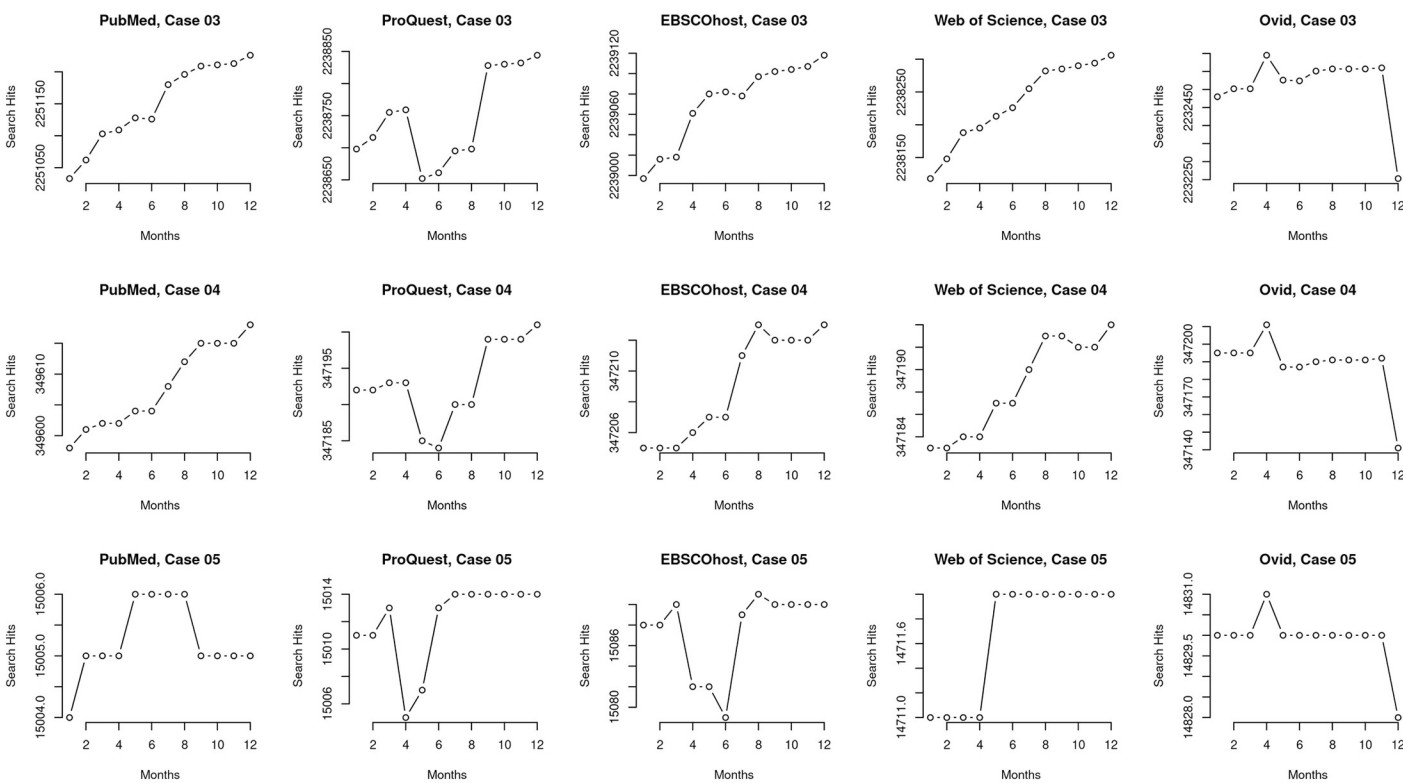

**Fig 4. Micro view of cases #03–#11.** A micro perspective of search result counts for cases #03–#11 that were restricted by publication date. Each row displays all five platforms individually within a case to aid side by side comparison of results. The micro perspective is able to show how the platforms vary in detail and trends; most importantly, they highlight different trajectories and differences in how platforms drop retrieved records even for publications with publication dates ending years before data collection.

results show that differences among MEDLINE platforms mainly show differences in search result counts but no or little differences in trends between the platforms for publications from 2015 or earlier based on queries to the systems in 2018 and later. Second, and in contrast, based on a detailed examination of the same cases (Figs 4–6), results show noticeable differences in trends between the platforms; that the bibliographic universe expands and contracts by dropping or adding records in 2018 and after for queries with publication dates ending in 2015 or earlier.

We show other database-based issues that hamper the reproduction of search strategies across MEDLINE platforms. These include the lack of comparable search fields in the studied MEDLINE platforms, an inconsistent pattern of results from some MEDLINE platforms that indicate a break in functionality, and a difference in lag time between updates from PubMed/ MEDLINE and other platforms in updating the database. External to these systems, we show how digital publishing complicates the bibliographic record, which was primarily designed in a print era. In the following sections, we describe the major themes of these differences among these platforms.

## Macro and micro views reveal different trends

We examined the data of the cases and present variations we identified based on two categories which we report in this section: an overview (macro) of the data (Fig 4) and a more granular (or, micro) view of the variations arising in the data (Figs 4–6).

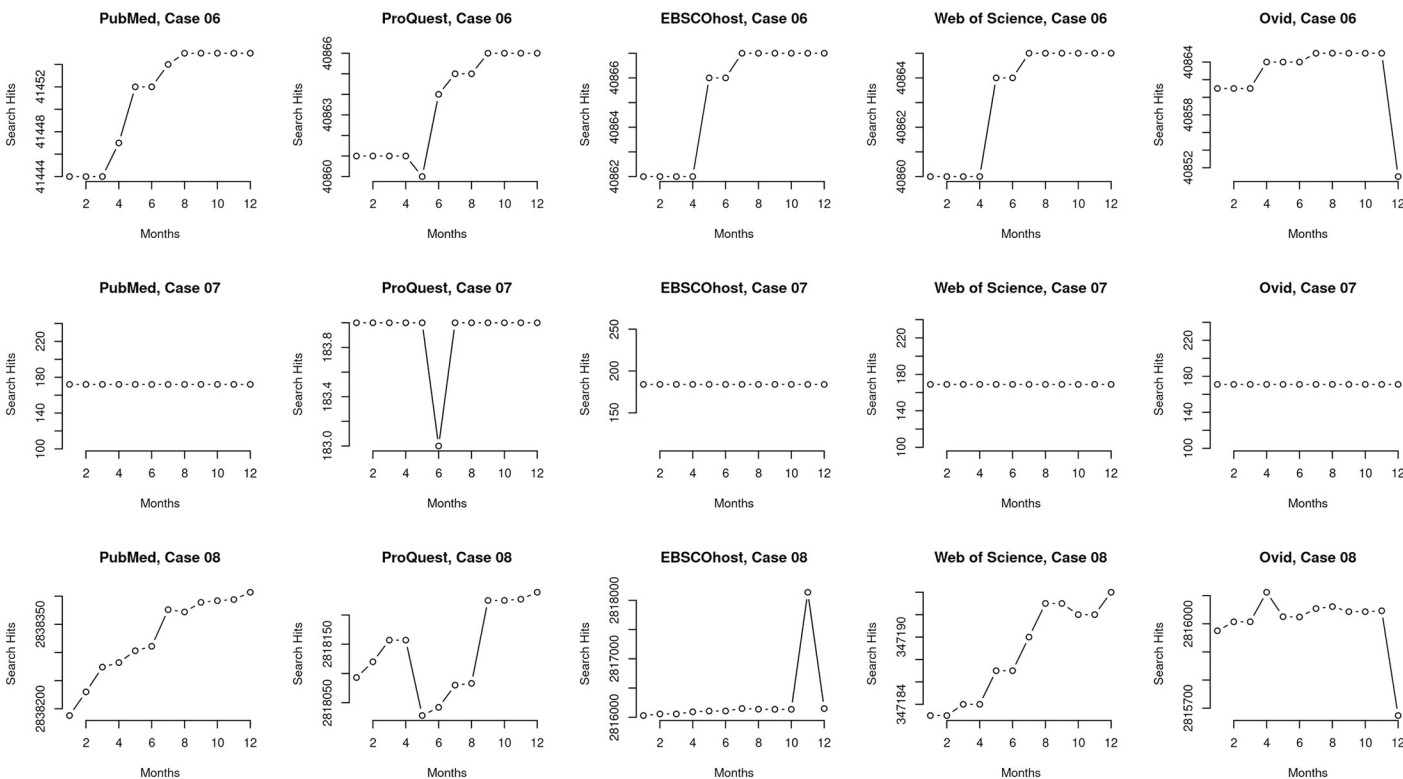

**Fig 5. Micro view of cases #03–#11.** A micro perspective of search result counts for cases #03-#11 that were restricted by publication date. Each row displays all five platforms individually within a case to aid side by side comparison of results. The micro perspective is able to show how the platforms vary in detail and trends; most importantly, they highlight different trajectories and differences in how platforms drop retrieved records even for publications with publication dates ending years before data collection.

The macro view of the cases restricted by publication date (cases #03–#11) mainly indicate that there are only differences in total search result counts over time among platforms within each set, for example, between WSML and PMML (Fig 3, top left plot) and that the trends are otherwise reliably comparable across time on a per-query, per-platform basis (Fig 3). For example, in case #05, which includes a query with a single MeSH term and an all-text keyword, all five platforms returned a range of 377 records in October 2018 and this increased by one to a range of 378 records in September 2019. However, the granular, micro view indicates variation within the platforms themselves (Figs 4–6) and that platforms are not internally reliable across time on a per-query basis—that they differ from each other at different dates and among each other at any given date. For example, case #05 (Fig 4) shows different search result patterns for each platform over the course of data collection. Even though this query is restricted by publication date, search results for each platform increase and decrease at different points in the year.

Figs 7 and 8 present the modified z-scores and highlight the deviations for all searches compared to PubMed, as the reference database, per case. Fig 7 includes all cases that were within 3.5± deviations from the range of search counts from PubMed. Fig 8 includes deviations outside that range and that are outliers. The figures highlight deviations not around the statistical center of the distribution of ranges but around the annual range of search counts from PubMed.

Fig 7 does not include statistical outliers but shows inconsistencies between PMML and the four platforms over the course of the year of data collection. PQML generally returned fewer

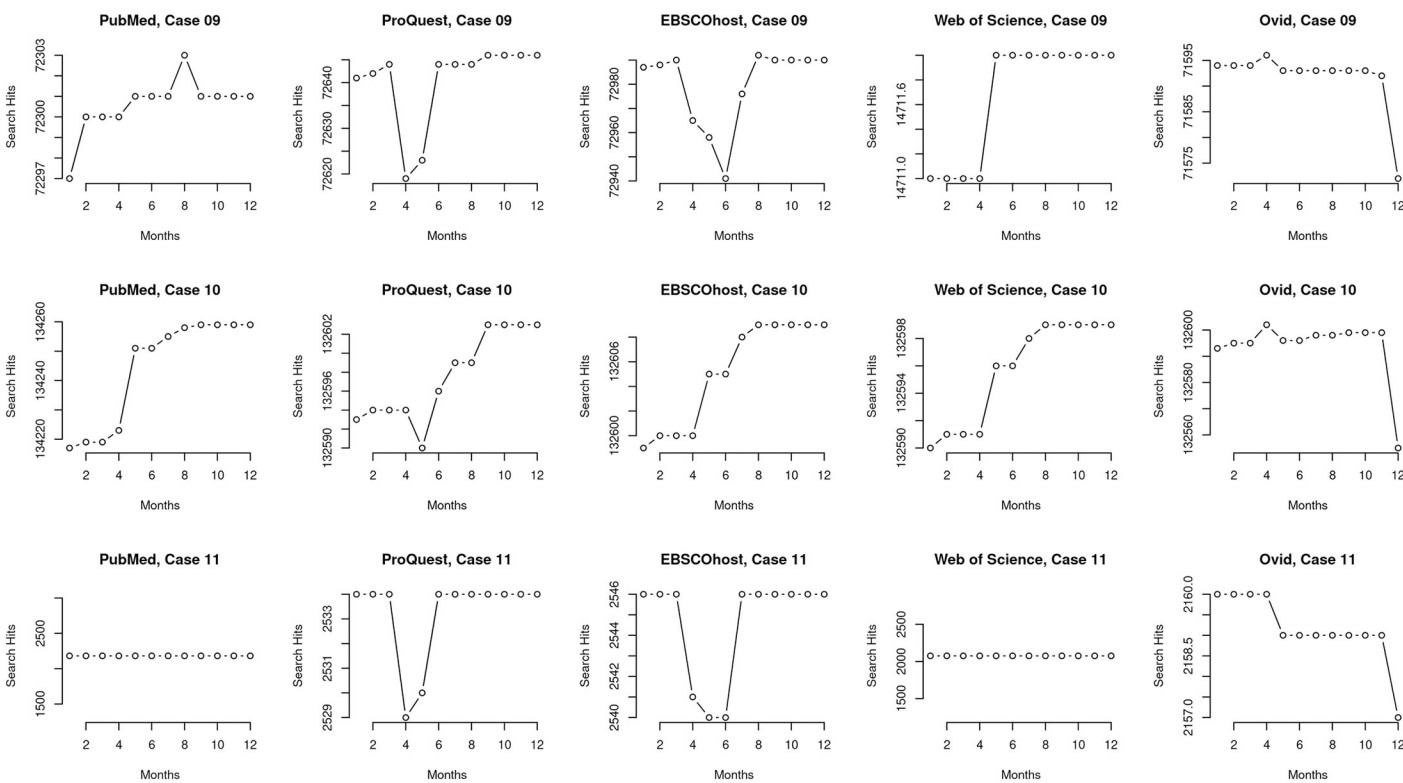

**Fig 6. Micro view of cases #03–#11.** A micro perspective of search result counts for cases #03–#11 that were restricted by publication date. Each row displays all five platforms individually within a case to aid side by side comparison of results. The micro perspective is able to show how the platforms vary in detail and trends; most importantly, they highlight different trajectories and differences in how platforms drop retrieved records even for publications with publication dates ending years before data collection.

records than PMML but returned more records for four cases, including two with publication dates in the queries. However, for WSML, cases with queries with publication dates returned fewer average records than PMML, and this is the opposite for OML. Fig 8 shows that all four non-PubMed platforms returned outliers, but out of the 16 cases present, only three appear more than once (Cases #03, #05, and #18) among all four platforms. Furthermore, several cases returned vastly different counts of results, compared to PMML. In Case #21, PQML consistently retrieved thousands more records than PMML for a query containing an all-fields keyword search and journal titles.

## Online-first versus issue-based publications impact bibliographic control

To help explain the changes in search results for queries that were restricted by publication dates, we compared query cases, #02 and #04, which were both designed to search for a single MeSH term ("neoplasms"), not exploded, but differed in that case #04 is publication date restricted. Hypothetically, searches for MeSH terms should not be impacted by changes in full-text search algorithms since the search process for a controlled term is based on whether a record contains the term or not in a well-defined search field. The grand median change over the year in search result counts for query #02 among all five platforms was 16,551 records (max: 17102; min: 15933), indicating the hypothetical annual growth in literature attached to this term since this query was not restricted by publication date. The grand median change in search results for query #04 among all five platforms was 17 records (max: 70; min: 8). Since

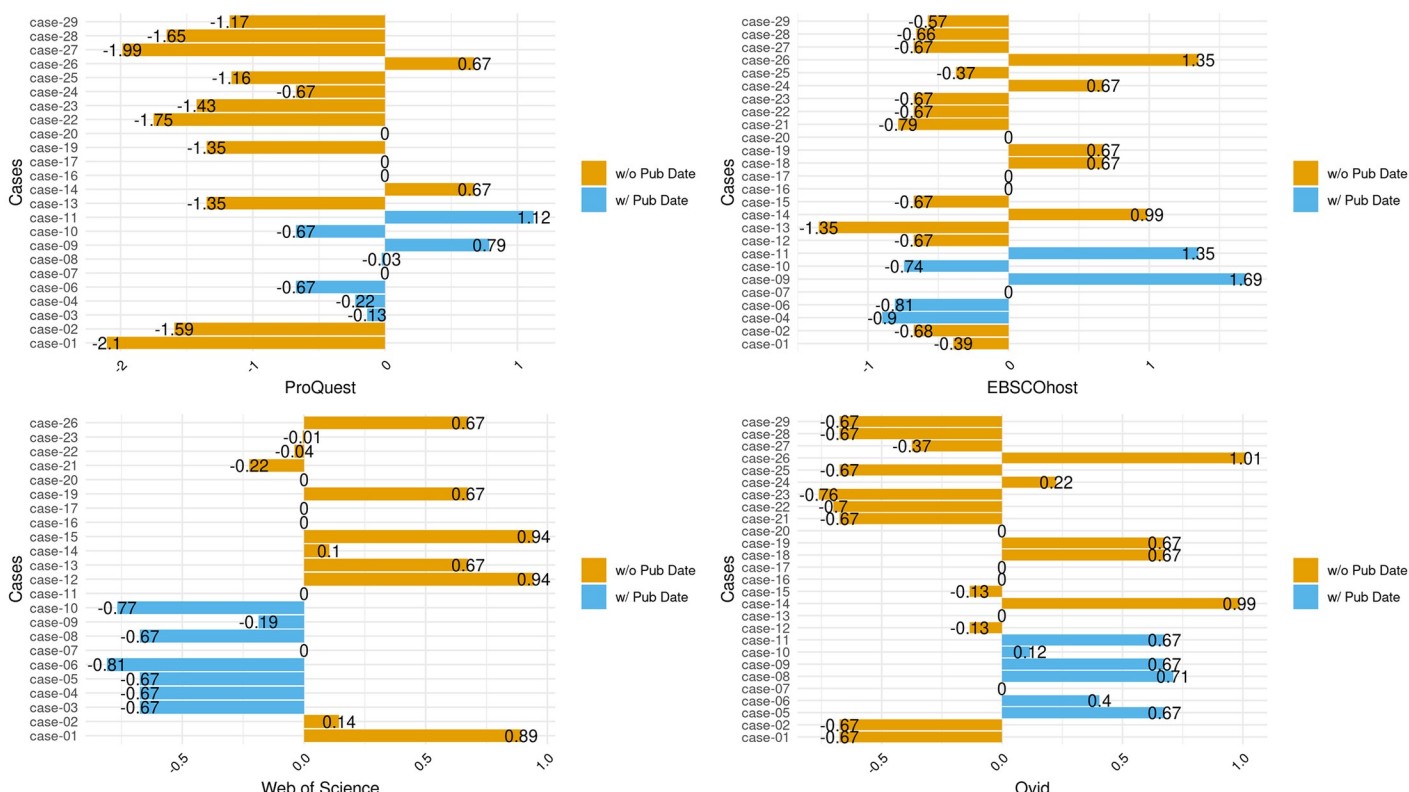

**Fig 7. Deviations per case and platform.** Deviations per case and platform with deviations centered around PubMed MEDLINE. Cases exclude outliers.

this query set was restricted by publication date, this suggests hypothetical changes that are not related to literature growth but to changes in either the database search indexes or the bibliographic records, four years after publication.

Furthermore, since all platforms reported different search result numbers in this query set, this indicates that the five platforms index different versions of the same MEDLINE file, or that the platforms index the same MEDLINE file differently based not on the MeSH term but on the publication date field since the growth cannot be explained by differences in how the platforms index MeSH fields. To test this, we traced a record from the query results for case #04 and investigated a top record that was part of the retrieval set for a query that was limited by the publication date to 2015 but which the record indicated it was published in 2019.

According to PubMed documentation, the default publication date field [DP] or [PDAT] includes the date of publication for either the electronic or print version [54]. An investigation of the chosen record from the search results for #04 in the PMML case [55] shows a bibliographic record with a multi-part publication history. The record indicates it was added to PubMed in 2015 but not formally entered into MEDLINE until 2019 (See: https://www.ncbi.nlm.nih.gov/pubmed/26700484?report=medline&format=text). On the journal (*BMJ*) article's web page, there are two versions of the article—an "Online First" version for the article that was issued in 2015 (See: https://spcare.bmj.com/content/early/2015/12/23/bmjspcare-2014-000835.info?versioned=true) and an "online issue" version of the article that was issued in 2019 (See: https://spcare.bmj.com/content/9/1/67.info). The journal article's publication history on its web page states that the 2015 version of the article is the "Online First" version, and the 2019 "online issue" is the version with a publication date for when the article was assigned

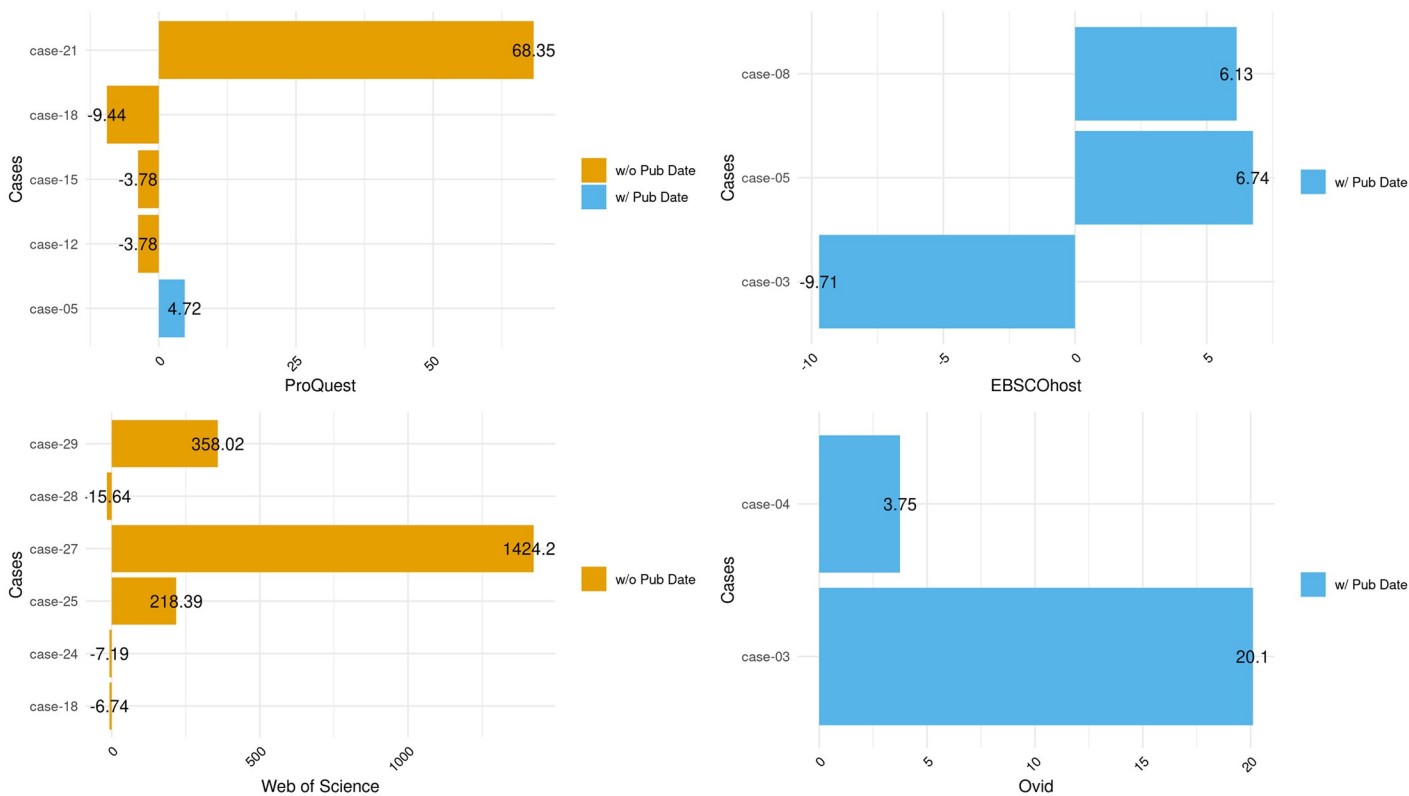

**Fig 8. Deviations per case and platform with outliers.** Deviations per case and platform with deviations centered around PubMed MEDLINE. Cases include outliers only.

to a volume and issue, and added to MEDLINE. On the journal's site, there are thus two versions of this article. However, on PubMed, there is one record for this article with two publication dates because of the versioning.

This record for the *BMJ* article above indicates problems with bibliographic control and dependency on journals to maintain bibliographic records that are complicated by two sequences of publications: online publications that precede publications attached to volume and issue numbers. The latter are complicated by the versioning of articles based on publication history and that include versions prior to their official publication dates when they are assigned to volume and issue numbers and then added to MEDLINE.

This problem with bibliographic control impacts search results across the platforms. The *BMJ* article described above does not appear in the search results among the other four MEDLINE platforms for case #04, and this confirms that the PubMed platform captures the electronic publication date by default even though the record was not entered into MEDLINE proper until the issue publication date. In this case, this was four years after the online publication date even though the query was restricted to MEDLINE. The other platforms do not behave in this way. The ProQuest and Web of Science MEDLINE-based platforms only offer the ability to delimit search by a single publication date type, which seems to be defined by the e-publication date. These platforms do not offer the ability to search by other date fields. The EBSCOhost and Ovid platforms offer more control over a variety of date fields but apparently the default publication date field is not inclusive of issue publication dates between these two platforms, like it is on PubMed.

## Reproducibility improved with field searching

We found that queries in cases that returned nearly equivalent search result counts were queries that included additional field specificity, regardless if the queries were restricted by publication date. Case #13 included two MeSH terms, the first one not exploded and the second one exploded, that were connected by one Boolean NOT, plus one document title term. All five platforms returned results that were within a range of 3 records, taking into account the range of results per platform and then among platforms over the annual period of data collection. This relative consistency across platforms was found in other search cases that included additional, specific field searches. For example, in case #18 we performed a single author search for an author who was chosen because they had published in journals indexed by MEDLINE during the middle part of the 20th century. The range of records that were returned varied over the months and numbered within a range of 15 records among the others. However, when a MeSH term was added to the author name search (case #17), chosen because the author had published papers that had been indexed with the specific MeSH term ("neoplasms"), all five platforms returned the same count of records for all twelve months of data collection.

## Database integrity

Aside from issues with bibliographic control due to online versioning, and with differences in indexing, several of the platforms returned results that appeared as outliers, or yielded vastly different search counts, compared to the other queries within their respective cases. The query in case #08 included one MeSH term, on a single branch, exploded, with a publication date restriction. PMML returned a range of 219 additional records across the months. PQML returned a range of 211 records, and OML returned a range of 438 records. However, EHML returned a range of 2108 records, and although WSML returned a range of only 11 records for the time period, it also returned an average of 2491138 fewer records than PMML. We could find no discernible reason for this discrepancy. Cases #10 and #23 both included a single MeSH term, two branches, exploded, and although search result counts were different among these platforms within these cases, the differences were not as extreme, perhaps then indicating a problem with how WSML explodes single branch MeSH terms.

There were two cases where one platform failed to return any results. In cases #24 and #28, WSML returned 0 results across all twelve months, even though the syntax of the queries was correct and one of the queries had returned results in a pilot test but then dropped them in subsequent tests. Additionally, in cases #25, #27, and #29, the WSML search result counts were initially within a reasonable range of the other platforms, but then diverged substantially (Fig 8). For example, in case #25, WSML returned a maximum of 13021 records and a minimum of 12652 records from October to April. However, the same query returned a maximum of 629 records and a minimum of 619 records from May to September, indicating a drop of over 12000 records for the same search query. In case #27, WSML returned search counts that were different but comparable to the other four databases, but then in May again, the counts increased by nearly 40000 records and remained within that range until the end of data collection. For case #29, the search result counts were again within range of the other four databases through April, but then in May and until the end of data collection, the query no longer retrieved any records. The only pattern among these three queries was that the sudden changes in search result counts occurred in May.

## Differences in database currency

The time it takes to import the overall PubMed file among platforms also impacts retrieval. On January 24, 2020, the US National Library of Medicine (NLM) recommended an interim

search strategy for searching all of PubMed for literature related to the COVID-19 virus [56]. Their initial recommended search query searched all fields for the term *2019-nCoV* or for the terms *wuhan* and *coronavirus* in the title and abstract fields (1)

$$2019 - \text{nCoV}[\text{All Fields}] \text{ OR } (\text{wuhan}[\text{tiab}] \text{ AND coronavirus}[\text{tiab}]) \qquad (1)$$

We modified the search strategy to use it among all platforms and queried PubMed, inclusive of MEDLINE by default, and the other platforms at irregular intervals. Results showed that for the PubMed data file, generally, all of the platforms returned different results for the same query for new literature. Results also included three versions of NLM's interface to PubMed. Two versions were for legacy PubMed but show result counts for when records were sorted by Best Match or Most Recent, since legacy PubMed applied two different sorting algorithms in this version of PubMed [30]. We also show search results for the new version of PubMed, which does not apply different sorting algorithms for Best Match or Most Recent, but which did report different search counts than both legacy PubMed results. As Fig 9 illustrates, these different platforms for PubMed retrieved newly added records at different rates, likely because they received and incorporated the PubMed data file at different times (Fig 9).

## Methods and results reproducibility

Overall, we found several issues that impact the unevenness of search results across these platforms and therefore with their use as reproducible instruments in scientific investigation. Due to differences in search fields across MEDLINE platforms, such as with the publication date and the MeSH fields, in developments in publishing, such as online first versions of articles versus volume and issue number versions, in the ability of databases to behave consistently

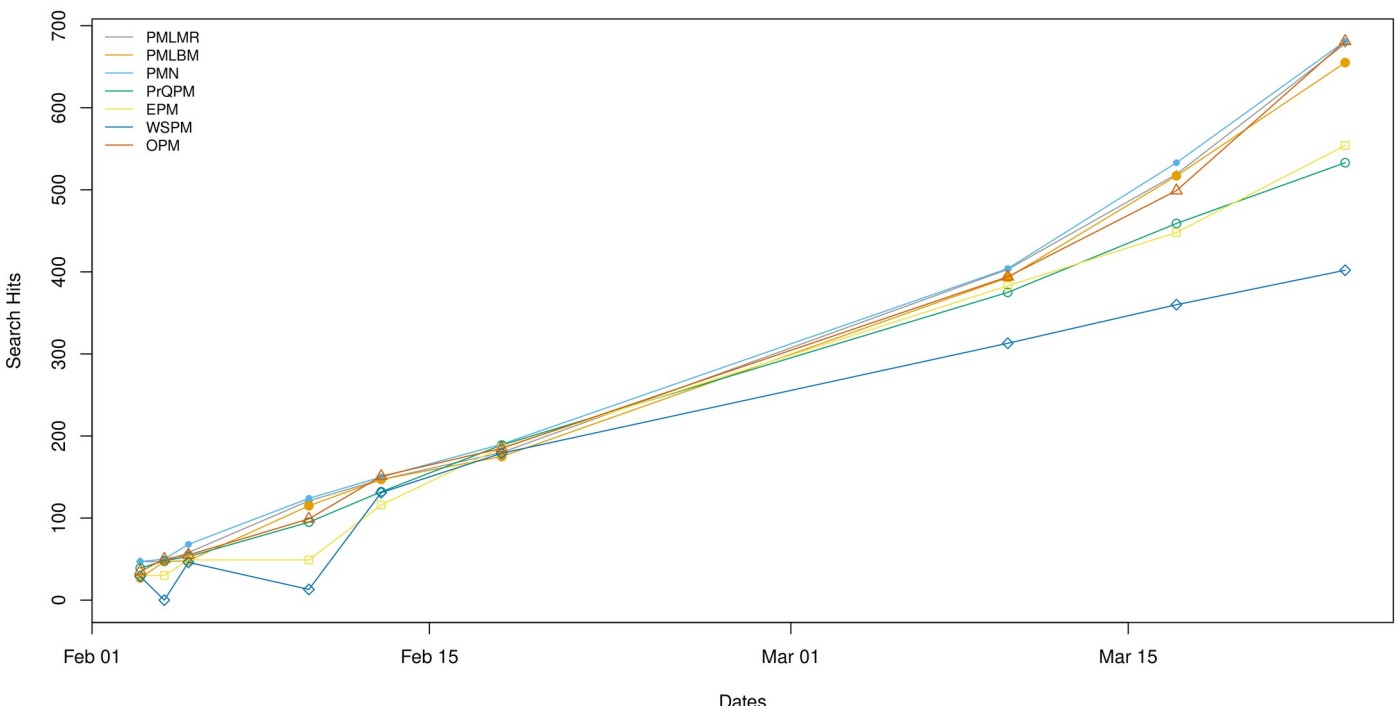

**Fig 9. Search result count differences for COVID-19 related searches across PubMed based platforms.** PubMed Legacy Most Recent Sort (PMLMR), PubMed Legacy Best Match Sort (PMLBM), PubMed New (PMN), ProQuest PubMed (PrQPM), EBSCOhost PubMed (EPM), Web of Science PubMed (WSPM), Ovid PubMed (OPM).

over time, and to differences in updates of the source file across platforms, it is difficult to construct queries that perform consistently alike across platforms and to get consistent results across them.

Specifically, we found that queries restricted by publication dates continue to return new records years past the limit on the publication date range. Data from this study provide some explanation for this variance. First, the growth of "online first" publications has complicated the traditional bibliographic record for journal articles which relies on the relationship between an individual article and its volume and issue. Additionally, although all platforms provide a search field or a way to limit search results by publication date, not all do so at the same level of detail. While simple publication date searching may have been sufficient in a print only age when there was only one publication date, it is not sufficient when articles are published multiple times, via electronic publication dates and via dates with volumes and issues. The implication is that records risk being dropped or added to time-restricted searches.

We found that queries were more likely to return comparable search result counts when they included multiple and specific field terms, such as queries combining keywords appearing in a journal title or an article title, with an author name, and a relevant MeSH term (e.g., cases #13 and #17). Practically speaking, this indicates that search results may be more uniform across platforms when searching for a known set of articles using a highly specific, multi-faceted search query. Conversely, simple queries using just one or two keywords or MeSH terms appear more susceptible to significant variations across platforms, underscoring the importance of advanced training in literature database searching and consultation with trained information professionals. However, future research studies could help identify how more complex queries might work more reliably across platforms.

Some platforms are not able to handle exploding the MeSH hierarchy similarly (e.g., EBSCOhost and Web of Science outliers in case #08), or they drop records from one month to the next even though the query has not altered. The lack of discernible causes in search result counts over time makes it impossible to adjust for such variance and undermines the trust in using bibliographic databases to inform data-driven decision making.

Our longitudinal study suggested that some differences might be attributed to delays in transferring the MEDLINE file to the vendors, since PubMed updates MEDLINE daily but the other vendors update at different times. To test this, we ran a COVID-19 search based on a query provided by the NLM in January 2020 and found that there were uneven search result hits for new literature on the COVID-19 pandemic across platforms. Although some of the differences in search result counts might be explained by the previous issues, the main explanation here is likely due to delays in receiving and incorporating the PubMed updates to the vendors. This suggests that if researchers need urgent access to the timely publications, they should be concerned about which version of PubMed they use to collect data.

## Limitations

One limitation of this study is that its comparison of search result counts does not provide any insight into the consistency of content retrieved across platforms. Without comparing specific papers' inclusion or omission per query, or how duplication affects retrieval, it is difficult to illustrate how retrieval inconsistencies might impact clinical care. Therefore, future research should examine the contents of retrieved works to better understand this dynamic.

Another limitation is that this research focuses on analyzing differences among permutations of relatively simple strategies. Future research should compare published systematic review or other advanced search queries to better understand the range of differentiation between simple and complex searches.

Further, the strategies developed for this study were purposive, but not comprehensive. With these search strategies, we intended to reflect many of the basic building blocks of searches that users may employ when conducting research in literature databases. However, the number of permutations a search strategy can have prevents a comprehensive approach. For instance, each term, field, limiter, and combination offers a new chain of outcomes to examine. Consequently, it would be beneficial for future research to include in-depth examinations of each variable across platforms.

## Future research

In previous work, and in the current study, we highlighted the importance of documenting variances in systematic searches across platforms based on the same data [49]. While systematic reviews often rely on complex combinations of fields and operators that are difficult, if not impossible to map accurately from platform to platform (take for example, the lack of proximity operators in NLM's PubMed interface), their use in health care decision making underscores the importance of examining variances across platforms based on the same data. If there are variances, why, to what extent, and further, what impact could those variances have on decision making? For example, we believe it is important to know whether justification is needed in choosing one MEDLINE-based platform over another when conducting searches and not assume that different MEDLINE-based platforms operate or return the same content based on semantically equivalent search strategies.

Though this study highlights significant retrieval differences across platforms, the remaining unknowns of how queries influence retrieval mean that offering concrete recommendations for end users is still difficult. However, queries using a combination of several metadata fields returned nearly equivalent search result counts, as compared to queries incorporating fewer metadata fields. Therefore, creating searches that incorporate several metadata fields when possible may lead to more consistent results than simpler searches. However, we must stress that more research is needed in order to understand how these databases function or malfunction. We think that the uncertainties that we raise about these systems, discovered by comparing them to each other, warrant some attention at standardizing the platforms, and that this is important given the critical role that literature retrieved from MEDLINE plays in the health and bioscience fields.

## Conclusions

MEDLINE is the National Library of Medicine's premier bibliographic database that contains more than 26 million references to journal articles in life sciences with a concentration on biomedicine. The subject scope of MEDLINE is biomedicine and health, broadly defined to encompass those areas of the life sciences, behavioral sciences, chemical sciences, and bioengineering needed by health professionals and others engaged in basic research and clinical care, public health, health policy development, or related educational activities. MEDLINE also covers life sciences vital to biomedical practitioners, researchers, and educators, including aspects of biology, environmental science, marine biology, plant and animal science as well as biophysics and chemistry [58].

Multiple vendors use the MEDLINE data to offer versions on their own platforms. In this study, we have found that issues that make reproducing searches across these platforms difficult, indicating that there is no one MEDLINE. Given the complexity in how various platforms index data fields (for example, automatic term mapping, proximity operators, date fields, and MeSH explosion), database providers should improve MEDLINE-specific documentation to aid instructors, information professionals, and perhaps to a lesser extent, novice end users, in

better understanding database function and implications of query construction. Since we show here that no MEDLINE-based platform is the same as any of the other platforms, specific platforms should be cited in any future research that uses one of these platforms for data collection. Alternatively, multiple MEDLINE-based platforms should be used when conducting searches, and all should be cited if multiple ones are used.

Remarkably, these results suggest we may be able to level one of the early critiques of Google Scholar, which was its inability to replicate results from the same search over periods of time [57], on MEDLINE. What followed with research on Google Scholar were several studies recommending against using Google Scholar as the sole database for systematic reviews [58, 59]. If this criticism is valid for the MEDLINE platforms, our results may strengthen the recommendation by Cochrane [22] that no single MEDLINE platform should be considered a sole source. We add that any platform that is used to gather bibliographic information be specifically cited in future research and that researchers enlist the assistance of library and information professionals when a comprehensive literature search is required. We also recommend that when multiple platforms are used, that researchers verify results across these platforms, but further research is needed to better understand generalizable patterns of differences across them, which admittedly may not be possible given constant changes in database technology.

The MEDLINE data is licensed to multiple vendors of information services who provide access to the database on their platforms, such as EBSCOhost, ProQuest, Ovid, and Web of Science. Any of these platforms are used by information specialists, health science librarians, medical researchers, and others to conduct research, such as systematic reviews, in the biomedical sciences. Our research examines results based on 29 cases that included 145 carefully constructed search queries, plus queries related to the COVID-19 pandemic, across these platforms and indicates that these platforms provide uneven access to the literature, and thus depending on the platform used, the validity of research based on the data gathered from them may be affected. Additional research is required to understand other search-related differences among these platforms, including differences among the records that are retrieved, and how they specifically impact research designs like systematic reviews and other biomedical research, and scientific conclusions based on these studies.

## Supporting information

**S1 Data.**
(XLSX)

## Author Contributions

**Conceptualization:** C. Sean Burns, Robert M. Shapiro, II, Jeffrey T. Huber.

**Data curation:** C. Sean Burns, Tyler Nix.

**Formal analysis:** C. Sean Burns.

**Investigation:** C. Sean Burns.

**Methodology:** C. Sean Burns, Robert M. Shapiro, II.

**Project administration:** C. Sean Burns.

**Resources:** C. Sean Burns.

**Software:** C. Sean Burns.

**Supervision:** C. Sean Burns.

**Validation:** C. Sean Burns.

**Visualization:** C. Sean Burns.

**Writing – original draft:** C. Sean Burns, Tyler Nix, Robert M. Shapiro, II, Jeffrey T. Huber.

**Writing – review & editing:** C. Sean Burns, Tyler Nix, Robert M. Shapiro, II, Jeffrey T. Huber.

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
