## [Decision Letter · Decision Letter 0]

7 Jul 2020

PONE-D-20-15022

Methodological Issues with Search in MEDLINE: A Longitudinal Query Analysis

PLOS ONE

Dear Dr. Burns,

Thank you for submitting your manuscript to PLOS ONE. After careful consideration, we feel that it has merit but does not fully meet PLOS ONE’s publication criteria as it currently stands. Therefore, we invite you to submit a revised version of the manuscript that addresses the points raised during the review process.

Please see the reviewers report at the end of this email. You will notice that they find merit in the paper but have raised some concerns regarding the design, methods and limitations of the study. They have also given some  suggestions to improve the paper.

We look forward to receiving your revised manuscript.

Kind regards,

Muhammad A. Z. Mughal

Academic Editor

PLOS ONE

Journal Requirements:

"I have read the journal's policy and the authors of this manuscript have the following

competing interests: Wolters Kluwer."

Reviewers' comments:

Reviewer's Responses to Questions

**Comments to the Author**

1. Is the manuscript technically sound, and do the data support the conclusions?

Reviewer #1: No

Reviewer #2: Yes

2. Has the statistical analysis been performed appropriately and rigorously? 

Reviewer #1: No

Reviewer #2: Yes

3. Have the authors made all data underlying the findings in their manuscript fully available?

Reviewer #1: Yes

Reviewer #2: Yes

4. Is the manuscript presented in an intelligible fashion and written in standard English?

Reviewer #1: Yes

Reviewer #2: Yes

5. Review Comments to the Author

Reviewer #1: This well-written paper presents results from a study comparing search results from the MEDLINE database in five retrieval systems. The study examined differences in the number of documents returned by each system for 29 cases. The study is motivated by the need to better understand discrepancies between the systems and changes in the results within each system. The study is novel in that it collected comparative data over a twelve-month period. The study was carefully designed to make results comparable between systems, with the focus on constructing Boolean queries for each case such that the syntax of each query was modified to conform to the coding and syntax of each system while duplicating the same criteria precisely. The authors also endeavored to construct a set of cases that allowed for a diversity of query functions and data structures between the cases. The authors have provided access to their extensive dataset. The paper contributes an overview of differences in the number of results returned and hypothesizes informally on some factors that may explain differences observed.

The goal of the study is reasonable, and the data collected are detailed and extensive, however the paper falls short in four important dimensions. First, the design of the study is not explained and justified sufficiently, making it hard for this reviewer to fully assess the validity of the design; this may be corrected by clearer explanation. Second, analysis of the voluminous data is insufficient in two major respects: (1) descriptive information, including between and within system differences, lacks statistics and (2) explanatory analysis of differences is ad hoc, with insufficient attention to supporting generalization on the results. These problems may be addressed by more detailed and systematic analysis of the data and by revision to the flow of the results section. Third, the novelty and importance of the findings and conclusions are unclear; this may be remedied by additional discussions of existing literature on the question of obtaining sufficient coverage for systematic reviews. Fourth, the limitations of the study are not addressed by the authors, a shortcoming that can be remedied by inclusion of discussion on this topic. My review covers these issues in turn, ending with some additional comments on specific passages.

THE DESIGN

The presentation of the design is confusing and would be strengthen by more detail. Readers would be helped by a simple diagram showing the five systems, the concept of what I call a “case” (see below), the 29 cases, and the monthly data collection. It would also be helpful to list and define the data you collected and the variable(s) you are analyzing from that data.

The term “sets” as an organizing device is confusing. The 29 sets are “cases”, where each case represents a specific set of subject term(s) and delimiters. Only when I reviewed the data in GitHub was I able to understand the structure of the cases, which is important to understanding the results. Readers need more information on the structure and motivation for the cases used in the paper. This may be provided by summarizing not only the search parameters (table 2) but the content values (e.g., subject terms, delimiter values in addition to date). In reviewing the data on GitHub I found that 28/29 cases cover only two topics (subject terms): neoplasm and dementia. The other case covers one author name. Each of the two topics was further qualified by publication, author, date, title, or related subject terms: immune, risk, and human. Motivation for the selection of these topic areas and the 29 cases are unclear. Why were these specific topics chosen? Why only two topics? What are the rationales for the specific variations in the construction of each case? This information is critical to your readers’ ability to assess the validity of your observations and conclusions. The key concern is that your design involves your direct control of the data you produced as your sample. For this reason you must be very explicit about the rationale for differences in the query constructions used in each case. Further, a systematic discussion of the similarities and differences in the 29 cases is essential because there is a great deal of overlap among the cases.

The very limited number of topics (subject terms) should also be explained. It has been long established that variability in search system results is highly dependent on the topic of searches used to compare systems (see list of references below). Generally, a comparative study with few diverse topics will yield biased results that misrepresent the average performance of a system over a given document collection. This is a limitation of the present study, thus the paper would be strengthened on why your approach is not biased by the two topics you chose. This goes back to the need to motivate your design more clearly. Better explanations are needed for why you used these specific topics and specific overlapping cases to compare the systems. You may also wish to compare your approach to other approaches such as using cases (topics/queries) from published systematic reviews.

Also, in explaining your rationale simplified language would help, and where esoteric concepts are involved (e.g., term trees, exploding terms) these must be carefully explained for unfamiliar readers. For unfamiliar readers, understanding *why* functionality matters is essential. The details presented in Table 2 are unhelpful because the effect of each parameter is not explained clearly to readers. This goes back to the need to justify your use of the parameters in the cases you constructed.

Finally, you state that you are not testing the performance of the systems (line 172-173: “the queries were not designed to test user relevance or user needs”) but “basic functionality”. With this rationale the concept of “basic functionality” must be well defined. It must be related directly to the specifics of your case design and to the measure(s) you are using for comparison. Without this, readers do not have enough information to assess the validity of your approach.

DATA ANALYSIS

You have elected to compare the number of items returned by each of your total 1,740 queries (29*12*5). No summary or descriptive statistics are provided on these counts. Results are presented in independent line graphs for each system, with data from only a subset of the cases presented. Your discussion of “macro” changes is difficult to follow (starts line 209). A simple statistical between-systems comparison of counts would be easier to follow. These could be averaged and/or presented in relation to the structure of your cases.

The graphical presentation of results is very unclear. First, the magnitude of differences over time, and between cases and systems is represented or discussed. Charts are drawn with inconsistent y-axes in figures 1 through 4, and no chart shows the zero value on this axis. Without discussion on the magnitude, the charts exaggerate the scale of changes and differences, although the trend-lines are clear. Trends and differences may be presented in statistical forms more easily communicated to readers. Data from all cases should be presented, or clear rationale for selected presentation must be given so your reader can better assess the validity of your analysis, results, and conclusions.

As importantly, your “micro-view” analysis would be greatly enhance by a more systematic explanation tied directly to the rationale(s) for case construction. As presented, these results read as ad hoc, with selected issues demonstrated by selection of specific cases. This reader is left with questions about whether the cases were constructed as demonstrations of specific issues known to exist in comparisons between MEDLINE sources (e.g., Duffy, de Kock, Misso, Noake, Ross, and Stirk, 2016). For this reason it is very important to provide a more systematic analysis of between and within-system differences. For example, within the cases presented there appear to be some similarities in trends which might be used to classify sources of discrepancies. Some trends appear to be simply growth in the indexed literature. Questions about similarity/difference in index growth could be analyzed by examining duplication of items in each list returned by each system. Other trends seem to be errors (incomplete index load) or error correction (adding back missing records). Again, detailed analysis comparing the index records returned could be used to evaluate the magnitude and scope of error-related trends, and how these compare between systems. After estimating the effects of these potentially large factors on differences between and within systems, more specific differences due to functionality might be explored. As currently presented the contribution and meaning of all these sources of variability are unclear.

Finally, you have elected to analyze only the quantity of items returned, and specifically to forgo analysis of results quality. The analysis would be greatly strengthened by an argument on why quantity is more important than quality, or at least, how quantity relates to quality.

MEANING OF RESULTS

Throughout the paper the question of “why” differences between systems matters is not completely clear. The paper would be greatly strengthened by discussion of the implications. Why do these findings matter? For whom do they matter? Are expert searchers aware of these differences? What do experts do to compensate? Evidence from the literature on how clinical decision-making, systematic review, knowledge bases are adversely impacted by differences would enhance readers’ understanding of the importance of these results. More specifically, how does each type of difference discussed relate to risks and adverse outcomes related to incomplete or erroneous information?

Importantly, some of the existing literature on differences between systems has not been cited. Please see the list below. These studies have found similar differences between and within systems. Relating your results to the prior findings would greatly strengthen the paper.

LIMITATIONS

The paper has no discussion on the limitations of the study. This is needed so that readers can understand how to apply the findings in a valid and trustworthy fashion.

Other comments:

Associate the example in table 1 with the information you provide in the text (lines 171- 181). Connect the example to the field names and explain how each relates to the concept of “functionality”

Drop table 2. The information is not useful to readers. More useful would be a more general explanation of why differences in these specific parameters matter and how they relate to the rationale for construction of your cases.

References on related findings:

Boeker M, Vach W, Motschall E. Semantically equivalent PubMed and Ovid-MEDLINE queries: different retrieval results because of database subset inclusion. J Clin Epidemiol. 2012 Aug;65(8):915–6. (See the paper discussed below). Note that the authors here describe a discrepancy created by an incorrect specification of an OVID query.)

Boeker M, Vach W, Motschall E. Time-dependent migration of citations through PubMed and OvidSP subsets: a study on a series of simultaneous PubMed and OvidSP searches. Stud Health Technol Inform. 2013;192:1196.

Duffy, S., de Kock, S., Misso, K., Noake, C., Ross, J., & Stirk, L. (2016). Supplementary searches of PubMed to improve currency of MEDLINE and MEDLINE In-Process searches via Ovid. Journal of the Medical Library Association: JMLA, 104(4), 309.

Duffy, S., Misso, K., de Kock, S., Noake, C., Ross, J., & Stirk, L. (2016, June). Supplementary searches of PubMed to improve currency of MEDLINE (Ovid) searches for systematic reviews: time taken for a record to move to MEDLINE. In 15th EAHIL Conference (pp. 6-11).

Kaunelis D, Farrah K, Severn M. Canadian Agency for Drugs and Technologies in Health. The missing 2%: PubMed NOT MEDLINE. Presented at: Canadian Health Libraries Association/Association des bibliotheques de la sante du Canada (CHLA/ABSC) 2010 Jun; ; Kingston, ON, Canada; 7–11.

Katchamart W, Faulkner A, Feldman B, Tomlinson G, Bombardier C. PubMed had a higher sensitivity than Ovid-MEDLINE in the search for systematic reviews. J Clin Epidemiol. 2011 Jul;64(7):805–7.

Lu, Z., Kim, W., & Wilbur, W. J. (2009). Evaluation of query expansion using MeSH in PubMed. Information retrieval, 12(1), 69-80.

US National Library of Medicine. Fact sheet: MEDLINE, PubMed, and PMC (PubMed Central): how are they different. [Internet] Bethesda, MD: The Library; 2016 [updated 6 Jan 2016; cited 2 Dec 2016]. < https://www.nlm.nih.gov/pubs/factsheets/dif_med_pub.html>.

Wanner, A., & Baumann, N. (2019). Design and implementation of a tool for conversion of search strategies between PubMed and Ovid MEDLINE. Research synthesis methods, 10(2), 154-160.

References on topic variability and system comparisons:

Banks, D., Over, P., & Zhang, N. F. (1999). Blind men and elephants: Six approaches to TREC data. Information Retrieval, 1(1-2), 7-34.

Robertson, S. E., & Kanoulas, E. (2012, August). On per-topic variance in IR evaluation. In Proceedings of the 35th international ACM SIGIR conference on Research and development in information retrieval (pp. 891-900).

Sakai, T. (2016). Topic set size design. Information Retrieval Journal, 19(3), 256-283.

Reviewer #2: I really enjoyed reading this manuscript. I appreciate the amount of work the researchers put into the project to collect data over a year with repeated searches in 5 bibliographic databases to track the search results. The analysis focused on identifying the inconsistent outputs and the causes of such issues across databases. The findings are important because missing publications can be life or death for a medical situation. The researchers also added the new topic CoViD-19 in the study, which is a plus.

I have only minor suggestions for consideration to help strengthen the manuscript:

In the Conclusion section, you wrote, "If this criticism is valid for the MEDLINE platforms, our results may strengthen the recommendation by Cochrane [23] that no single MEDLINE platform should be considered a sole source." Although you are hedging the findings and advice, I still hope to hear more than echoing Cochrane. Are there search strategies you would recommend for searchers to handle specific databases and cross database validation of search results? Further, what improvements the database providers should make? I think an important outcome of your results is to guide also practice in addition to understanding the phenomena.

I wondered if Figure 5 could have the white and black bars for the same search set side-by-side (i.e., the two bars next to each other) and also add data labels for easy visual comparison.

editing:

(The paper reads well-written. These perhaps more a stylist preferences by different readers.)

line 57, change "full text sources"  full-text sources

Line 116, "This is commonly known problem" [add "a" before commonly]

Line 163, "on a per set basis"  on a per-set

Line 182 - 185 "We added date ranges to test whether search result counts would remain frozen after the end of the publication date and to add an additional control that would help explain differences between the platforms." would read clearer if rephrased or illustrated with the actual queries.

Line 189, "field specific keywords"  field-specific keywords

Line 190, the same need to hyphenate the two words field-specific

Line 217, "on a per query basis"  on a per-query basis

Peiling Wang (UTK)

6. PLOS authors have the option to publish the peer review history of their article (what does this mean?). If published, this will include your full peer review and any attached files.

Reviewer #1: No

Reviewer #2: **Yes: **Peiling Wang

---

## [Author Response · Author response to Decision Letter 0]

20 Aug 2020

Response to reviewer file is attached.

---

## [Decision Letter · Decision Letter 1]

12 Oct 2020

PONE-D-20-15022R1

Methodological Issues with Search in MEDLINE: A Longitudinal Query Analysis

PLOS ONE

Dear Dr. Burns,

Thank you for submitting your manuscript to PLOS ONE. After careful consideration, we feel that it has merit but does not fully meet PLOS ONE’s publication criteria as it currently stands. Therefore, we invite you to submit a revised version of the manuscript that addresses the points raised during the review process.

As you can see in the review report at the end of this email. Both reviewers agree that it is a significantly improved version; however, they have raised some issues regarding the presentation of results, literature, and what could be the recommendations/application of these findings for MEDLINE or other similar vendors/databases. These comments would be helpful while revising the manuscript.

We look forward to receiving your revised manuscript.

Kind regards,

Muhammad A. Z. Mughal, PhD

Academic Editor

PLOS ONE

Reviewers' comments:

Reviewer's Responses to Questions

**Comments to the Author**

1. If the authors have adequately addressed your comments raised in a previous round of review and you feel that this manuscript is now acceptable for publication, you may indicate that here to bypass the “Comments to the Author” section, enter your conflict of interest statement in the “Confidential to Editor” section, and submit your "Accept" recommendation.

Reviewer #1: (No Response)

Reviewer #3: (No Response)

2. Is the manuscript technically sound, and do the data support the conclusions?

Reviewer #1: Yes

Reviewer #3: Partly

3. Has the statistical analysis been performed appropriately and rigorously? 

Reviewer #1: Yes

Reviewer #3: I Don't Know

4. Have the authors made all data underlying the findings in their manuscript fully available?

Reviewer #1: Yes

Reviewer #3: Yes

5. Is the manuscript presented in an intelligible fashion and written in standard English?

Reviewer #1: Yes

Reviewer #3: Yes

6. Review Comments to the Author

Reviewer #1: Second review

The authors have made substantial and helpful changes to the paper, which greatly improve readability and clarity. The materials and methods section is greatly improved with the additional explanation of the queries and the two diagrams. The new limitations and future research sections are excellent. Below I’ve mentioned some important changes to further enhance clarity and a section that needs an extensive revision (lines 229 through 274).

You’ve used the term “sets” to mean different things. Clarifying further with some minor changes would make it easier to follow. Where you are referring to results returned by a query call these “results sets” (or retrieval sets, but be consistent). Where you are referring to the queries comprising a case, call these “query sets”. There are a lot of places where “sets” is confusingly ambiguous. Being much more specific about what you are referencing with the term “sets” would make the paper easier to read and understand. I suggest that where possible you drop the term “set” and say specifically what you are talking about. See specific suggestions below.

The initial presentation of results is still very difficult to read and follow. This is an important part of the paper and it needs to be much clearer, particularly when you refer to comparisons such as “within”, “between” and “over”.

You are using Macro and Micro to organize the presentation, but these are not defined for the reader and the terms are not informative. Use your intro paragraph to explain what you will be presenting as results. As I understand it from the abstract, there are four main findings. WITHIN results from each individual platform, online-first publishing creates metadata differences that cause changes in results sets over time for a single work. Three causes are found for differences in results sets BETWEEN platforms: 1) differences in search field specificity causes differences between platforms; 2) differences in the timing of file updates causes differences between platforms; and 4) general data integrity problems cause differences between platforms. Presenting the results as being about either changes over time WITHIN a platform, or changes over time BETWEEN platforms, would make it much easier to understand.

Your first two sentences in the results section are extremely hard to understand: (line 238) “A macro examination of the search cases restricted by publication date (cases #03–#11) mainly indicate that there are only substantial differences in total search result counts over time between platforms within each set, for example, between WSML and PMML (Figure 3, top left plot). This macro view indicates that although there are differences among platforms with respect to absolute differences in search result counts, the trends are reliably comparable across time on a per-query, per-platform basis (Figure 3).” I’ve read through this several times and tried, but I do not understand your point.

The multitude of charts (which are so small they are nearly indecipherable) are not helping, nor is the rest of the paragraph. The heading for Figures 3 and 4 are not informative. The headings need to tell the reader what they are looking at. Because the legend for Figure 3 cannot be read (it is extremely blurry) the whole figure is not useful and it creates confusion, particularly when compared with Figure 4. Figure 4 seems to show the same data (counts by platform over time for separate cases), but in F3 the counts appear to be constant while in F4 the counts vary. Readers need a clear explanation, in simple language, of what is depicted in both figures, why they are different, and why it matters. I still believe you do not need most of your figures, particularly those that are truly unreadable. The information in Figures 3, 4, and 5 could be best represented with a small set of exemplars for each type of trend.

I suggest you take the reader through one full example of a case and explain each figure as you go. Generally, lines 229 through 274 should be revised extensively. As food for thought, your rebuttal comments are much clearer than the paper. There you have written in clear, more direct language that communicates your points more effectively.

Examples of ambiguity in the use of the term “sets”

In the abstract: line 17 “We found that search results vary considerably depending on MEDLINE platform, both within sets and across time.” ”… within sets …” is very unclear. Do you mean “… within results sets….” or “within cases” or something else?

Line 37 “queries influence information retrieval sets were once common lines of inquiry” Clarify that this as “retrieval results sets”.

Line 127 “…. influence different search behaviors and retrieval sets [45,46].” Clarify that this as “retrieval results sets”.

Line 132 “… evaluating recall and precision on the retrieval sets in these systems”. Clarify that this as “retrieval results sets”

Line 172 “…equivalent to each other on a per set basis.” Clarify this as “equivalent to each other on each platform. Throughout the paper we refer to the equivalent queries as query sets (italics).”

Line 180 Table 1 refers to “Search Set #” . Retitle the column as “query set #”. The whole table should be labeled something like “Example Case showing semantically equivalent queries for the five platforms”.

Line 202 “…in order to compare search count results based on logical sets.” This is unclear. As I read it, you wanted to “…compare each platform’s Boolean processing”

Line 214 “…a comparison of search result counts within each set”. This confusing. “Within sets” obscures the fact that you are comparing the platforms. You are comparing BETWEEN the results sets of each platform for each case.

Line 229 “Our query sets include 100 queries without publication date ranges and nine sets containing..” Clarify as “… and nine cases containing ….”

Line 234 “…the differences we found across the search sets,…” Clarify – are these differences found BETWEEN platforms or differences found WITHIN platforms?

Line 337 “This relative consistency across platforms was found in other search sets that included additional…” Clarify as “…was found in other cases that included…”

Line 355 “…although search result counts were different among these platforms within these sets,…” Clarify as “among these platforms for these cases….”

Line 363 “…initially within a reasonable range of the other platforms in the respective sets, but then..” Clarify by simplifying and saying “within a reasonable range of the other platforms, but then…”

Reviewer #3: My comments to the authors are as follows:

- As a biomedical librarian. I experiment regularly with the search functionality of many platforms.

- I also have performed complex search queries on the platforms used in your query analysis (PubMed, ProQuest, EBSCOhost, Web of Science, and Ovid) and have made cross comparisons over the years.

- I can tell you that these problems are not new; however, I concur anecdotally with what you have proven in your manuscript and your year long + longitudinal analysis. I did not spend any time checking the accuracy of the statistics.

- In using those platforms, I have also observed that controlled vocabulary searches across PubMed, EBSCOhost & Ovid are mostly similar / consistent in terms of retrieval numbers (especially back dating) and that those searches can be translated into the different syntax quite reliably. Searches involving current publication years and current annual indexing always produce varying numbers.

- Further, any type of searching involving author supplied keyword fields, searching in the titles and abstract fields, or any type of adjacency and frequency searching introduces varying and confusing numbers in all platforms. As you know, PubMed for example does not permit proximity / frequency searching although there is a rumour going around on Twitter that it is possible in the new PubMed interface.

- Given your paper's findings and what it adds to the body of evidence in searching Medline, what is your recommendation to searchers? You don't seem to address this at all in your conclusion.

- How should this evidence inform our work? Should we simply expect different results as we move across platforms and remember to watch for these in performing simple searches? Is that the message? I can tell you with certainty that most searchers are aware of this type of deficiency and how the problems shift over time. In other words, it's not news and the problems you uncovered are not stable and will be different next year.

- Further to your findings, I'm unclear about generalizability to other search queries. You mention that, in creating your search queries, you've given attention to queries that are semantically similar overall. I'm not clear what you mean; are these search queries generalizable to other sets of queries, and as such do they reveal broader patterns for each of the platforms? Further, would you expect to find similar patterns and results if you tested a different set of 125 queries?

- Can I extrapolate from your findings that there is, thank goodness, more consistency across Medline platforms when the search queries are complex (for example, during multi-set systematic review searching)? If so, that would be worth emphasizing or at least surmising yes or no in the conclusion for the simple reason that it will reassure systematic searchers.

- I read a similar paper you authored published last year in the Journal of the Medical Library Association: Burns CS, Shapiro RM, II TN, Huber JT. Search results outliers among MEDLINE platforms. Journal of the Medical Library Association: JMLA. 2019 Jul;107(3):364.

- In comparing the two papers, apart from testing the searches over time, how much of the concept and analysis here are expanded upon and/or different from your 2019 paper? You do allude to similar work you have performed in the past early in your manuscript but you don't discuss this paper specifically unless I missed it. The titles of the two papers are different; are they, in your view, sufficiently different / distinct both in terms of conceptual design and methods to warrant two separate manuscripts? If they are connected, by virtue of concept or overall goals for your research, you should say so.

- Further, the JMLA paper in 2019 is well-written though the PLoS manuscript is more detailed and of course deals with the longitudinal aspects. The title of the 2019 manuscript describes the paper accurately; it would be worth adapting that title to your PLoS manuscript, using the same economy of words and approach, to connect the two papers. The current title does not in my view convey the subject content of the manuscript in a descriptive way.

- In comparing your two efforts, I'm aware of added sections in the current manuscript pertaining to calendar year (2020) with regard to searches for COVID-19. That is timely and potentially useful but the question is still whether this adds substantively to our understanding.

- On line 396, you indicate that Medline "contains more than 25 million references to journal articles" but this number is by now in 2020 considerably higher than that, and possibly now as high as 28 million.

- For this review, I have had the privilege of being able to read the previous peer reviewers' comments of your paper, and your responses.

- I agree with the previous peer reviewers' comments generally but in particular want to emphatically agree with the comments / question pertaining to how you devised the search queries. It's unclear to me, even after reading the methods section again, why you chose to focus on the queries you did as a basis for testing consistency. Did the medical librarian on the team provide insight? I keep going back to comments earlier in this review in that I would have thought truly complex search queries would be more revealing.

- The focus on quantity of search results / retrieval numbers over any substantive discussion of quality of those results ("what exactly did you find when the numbers didn't agree across platforms? why did those extra references / citations appear in the retrieval sets?") is a major deficiency in the paper (which you acknowledge in your limitations section).

- In an ideal manuscript, it would be advisable to have a balance of both quantitative and qualitative aspects of your query analysis; perhaps your research team could have waited to publish until that part of your work was completed as you mention this paper is meant to be the start of a larger evaluation process.

- What do I tell my students and other expert searchers/ users who might read this paper? I hope you are not suggesting we search multiple Medline platforms to compare results ourselves for simple search queries, or to look for additional papers that the individual platforms were unable to find.

- Finally, as a medical librarian and information retrieval expert, what does this paper tell me and how does it inform my work? In the final analysis, it is not clear to me what your message is - other than to watch for variable results in simple searches on the various platforms. Perhaps you should send your manuscript to all the vendors for comment and so they can fix the problems.

- Best of luck.

7. PLOS authors have the option to publish the peer review history of their article (what does this mean?). If published, this will include your full peer review and any attached files.

Reviewer #1: No

Reviewer #3: **Yes: **Dean Giustini

---

## [Author Response · Author response to Decision Letter 1]

12 Nov 2020

Reviewer #1: Second review

The authors have made substantial and helpful changes to the paper, which greatly improve readability and clarity. The materials and methods section is greatly improved with the additional explanation of the queries and the two diagrams. The new limitations and future research sections are excellent. Below I’ve mentioned some important changes to further enhance clarity and a section that needs an extensive revision (lines 229 through 274).

You’ve used the term “sets” to mean different things. Clarifying further with some minor changes would make it easier to follow. Where you are referring to results returned by a query call these “results sets” (or retrieval sets, but be consistent). Where you are referring to the queries comprising a case, call these “query sets”. There are a lot of places where “sets” is confusingly ambiguous. Being much more specific about what you are referencing with the term “sets” would make the paper easier to read and understand. I suggest that where possible you drop the term “set” and say specifically what you are talking about. See specific suggestions below.

Thank you for this suggestion. We agree. Instead of globally replacing ‘sets’ with ‘query sets’ or ‘results sets’, though, we went through the manuscript and removed some instances of the term, replaced some with the term ‘cases’, where appropriate and based on the suggestion in a prior review, replaced the term with other wording, as appropriate, and sometimes simply removed the term and modified the sentence in other ways. We believe the manuscript is clearer now.

The initial presentation of results is still very difficult to read and follow. This is an important part of the paper and it needs to be much clearer, particularly when you refer to comparisons such as “within”, “between” and “over”.

You are using Macro and Micro to organize the presentation, but these are not defined for the reader and the terms are not informative. Use your intro paragraph to explain what you will be presenting as results. As I understand it from the abstract, there are four main findings. WITHIN results from each individual platform, online-first publishing creates metadata differences that cause changes in results sets over time for a single work. Three causes are found for differences in results sets BETWEEN platforms: 1) differences in search field specificity causes differences between platforms; 2) differences in the timing of file updates causes differences between platforms; and 4) general data integrity problems cause differences between platforms. Presenting the results as being about either changes over time WITHIN a platform, or changes over time BETWEEN platforms, would make it much easier to understand.

Thank you for this suggestion. We have revised the intro to clarify what we mean by macro and micro and why we are using these two perspectives to look at the data.

Your first two sentences in the results section are extremely hard to understand: (line 238) “A macro examination of the search cases restricted by publication date (cases #03–#11) mainly indicate that there are only substantial differences in total search result counts over time between platforms within each set, for example, between WSML and PMML (Figure 3, top left plot). This macro view indicates that although there are differences among platforms with respect to absolute differences in search result counts, the trends are reliably comparable across time on a per-query, per-platform basis (Figure 3).” I’ve read through this several times and tried, but I do not understand your point.

We have revised the intro to the Results section and have added a short explanation of the macro and micro perspectives and what they have to offer.

The multitude of charts (which are so small they are nearly indecipherable) are not helping, nor is the rest of the paragraph. The heading for Figures 3 and 4 are not informative. The headings need to tell the reader what they are looking at. Because the legend for Figure 3 cannot be read (it is extremely blurry) the whole figure is not useful and it creates confusion, particularly when compared with Figure 4. Figure 4 seems to show the same data (counts by platform over time for separate cases), but in F3 the counts appear to be constant while in F4 the counts vary.

This is the key point of these charts. They show the same data but at different views, and it is these different views which are one of the main findings of this paper. We have altered and revised the text to make the difference between the macro and micro views clearer and to highlight their role in this paper. Fig 3 shows that these platforms do not appear all that different when looking at them from up high, or macro, but Fig 4 shows that when the same data is examined up close, they don’t trend the same way, even though each panel should, theoretically, be exactly the same.

We are not sure why the charts are reading as indecipherable. They conform to the plot specifications required by PLOS ONE and we uploaded files with the resolution of 2250 x 1266 pixels. They are not blurry on our systems. All we can do is double check with the manuscript submission process that the files are rendered correctly. However, we agree that the legend for Fig 3 was too small and not readable enough, and we have fixed that.

Readers need a clear explanation, in simple language, of what is depicted in both figures, why they are different, and why it matters. I still believe you do not need most of your figures, particularly those that are truly unreadable. The information in Figures 3, 4, and 5 could be best represented with a small set of exemplars for each type of trend.

We think these plots are important and central to the main findings of this paper. However, we have revised the Figure captions and added more detail about what the plots convey, and this detail is influenced by your suggestion above about the micro and macro views. We have also replotted the data to include better legends.

I suggest you take the reader through one full example of a case and explain each figure as you go. Generally, lines 229 through 274 should be revised extensively. As food for thought, your rebuttal comments are much clearer than the paper. There you have written in clear, more direct language that communicates your points more effectively.

Thank you for this suggestion. We have heavily revised this part of the manuscript. We borrowed some of the language from our rebuttal and more completely introduced and framed the results.

Examples of ambiguity in the use of the term “sets”

In the abstract: line 17 “We found that search results vary considerably depending on MEDLINE platform, both within sets and across time.” ”… within sets …” is very unclear. Do you mean “… within results sets….” or “within cases” or something else?

We have revised this.

Line 37 “queries influence information retrieval sets were once common lines of inquiry” Clarify that this as “retrieval results sets”.

We fixed this.

Line 127 “…. influence different search behaviors and retrieval sets [45,46].” Clarify that this as “retrieval results sets”.

Fixed.

Line 132 “… evaluating recall and precision on the retrieval sets in these systems”. Clarify that this as “retrieval results sets”

Fixed.

Line 172 “…equivalent to each other on a per set basis.” Clarify this as “equivalent to each other on each platform. Throughout the paper we refer to the equivalent queries as query sets (italics).”

Fixed, but since we are using the word cases now, we have modified the above to reflect that.

Line 180 Table 1 refers to “Search Set #” . Retitle the column as “query set #”. The whole table should be labeled something like “Example Case showing semantically equivalent queries for the five platforms”.

Fixed.

Line 202 “…in order to compare search count results based on logical sets.” This is unclear. As I read it, you wanted to “…compare each platform’s Boolean processing”

Fixed.

Line 214 “…a comparison of search result counts within each set”. This confusing. “Within sets” obscures the fact that you are comparing the platforms. You are comparing BETWEEN the results sets of each platform for each case.

Fixed.

Line 229 “Our query sets include 100 queries without publication date ranges and nine sets containing..” Clarify as “… and nine cases containing ….”

Fixed.

Line 234 “…the differences we found across the search sets,…” Clarify – are these differences found BETWEEN platforms or differences found WITHIN platforms?

Fixed as part of the major revision to this section.

Line 337 “This relative consistency across platforms was found in other search sets that included additional…” Clarify as “…was found in other cases that included…”

Fixed.

Line 355 “…although search result counts were different among these platforms within these sets,…” Clarify as “among these platforms for these cases….”

Fixed.

Line 363 “…initially within a reasonable range of the other platforms in the respective sets, but then..” Clarify by simplifying and saying “within a reasonable range of the other platforms, but then…”

Fixed.

Reviewer #3: My comments to the authors are as follows:

- As a biomedical librarian. I experiment regularly with the search functionality of many platforms.

- I also have performed complex search queries on the platforms used in your query analysis (PubMed, ProQuest, EBSCOhost, Web of Science, and Ovid) and have made cross comparisons over the years.

- I can tell you that these problems are not new; however, I concur anecdotally with what you have proven in your manuscript and your year long + longitudinal analysis. I did not spend any time checking the accuracy of the statistics.

We have heard the same anecdotal stories and share similar experiences. The motivation for this study was to document these issues in order to have strong evidence for the problems and not just anecdotal support. Your comment below is noted about the importance of strengthening suggestions for searchers, but we also think the problem 1) lies with the vendors, their documentation, their implementation of MEDLINE, and more, 2) publishers who use multiple publication dates that only serve to complicate the bibliographic record simply to have articles assigned to volume and issue numbers someday down the line, and 3) the lack of attention paid to this problem and the need for more studies not just by us but by others in the field. We think the issues with these databases that we report here could raise serious, clinical problems for data based on items retrieved from these databases, including systematic reviews. We have added some text or strengthened some of our recommendations to help clarify this.

- In using those platforms, I have also observed that controlled vocabulary searches across PubMed, EBSCOhost & Ovid are mostly similar / consistent in terms of retrieval numbers (especially back dating) and that those searches can be translated into the different syntax quite reliably. Searches involving current publication years and current annual indexing always produce varying numbers.

Yes, we hope that this helps document inconsistencies and consistencies among MEDLINE platforms.

- Further, any type of searching involving author supplied keyword fields, searching in the titles and abstract fields, or any type of adjacency and frequency searching introduces varying and confusing numbers in all platforms. As you know, PubMed for example does not permit proximity / frequency searching although there is a rumour going around on Twitter that it is possible in the new PubMed interface.

Yes, again, we think this has been poorly documented when MEDLINE platforms are compared to each other. 

- Given your paper's findings and what it adds to the body of evidence in searching Medline, what is your recommendation to searchers? You don't seem to address this at all in your conclusion.

- How should this evidence inform our work? 

Honestly, we do not think we know enough yet to make a recommendation like this. In our conclusion, we recommend potential research projects that need to be undertaken. We seriously believe that a comparison among MEDLINE systems is understudied but that it reveals problems not just between systems but in each system. More complex queries, as you comment below, would be worthwhile, but this baseline documentation is important, we think, where we start to measure how these platforms behave with very simple queries. The more complex the query, as we note in the paper, the more difficult it will be to track problems in these systems.

Should we simply expect different results as we move across platforms and remember to watch for these in performing simple searches? Is that the message? I can tell you with certainty that most searchers are aware of this type of deficiency and how the problems shift over time. In other words, it's not news and the problems you uncovered are not stable and will be different next year.

Again, it’s not well documented. We also agree that many expert searchers are aware of these problems anecdotally. We also have the kinds of personal experiences with these platforms, which is what motiviated this study. But until this is scientifically documented, as we have tried to do here and in our JMLA paper, anecdotal evidence and personal experience are scientifically insufficient, and thus have little chance to support change in these systems, however that change might look once these issues are sorted out more. We do think that such change might involve some standardization and that this might be aided by standards organizations like NISO as well as the creators of these systems, like NLM, and other stakeholders, like librarians, medical professionals, and more.

- Further to your findings, I'm unclear about generalizability to other search queries. You mention that, in creating your search queries, you've given attention to queries that are semantically similar overall. I'm not clear what you mean; are these search queries generalizable to other sets of queries, and as such do they reveal broader patterns for each of the platforms? Further, would you expect to find similar patterns and results if you tested a different set of 125 queries?

We do not think that generalizability is important here in the standard inferential sense. What we think is important here is that our confidence in these systems is not well warranted if they behave as inconsistently as we have documented with these 29 cases of queries and after 12 months of data collection. We are also not sure if generalizability is possible here. Even if we detect patterns in these inconsistencies between platforms, database technology changes, scholarly publishing changes, and more. There is simply too much randomness to infer from specific patterns over time. 

- Can I extrapolate from your findings that there is, thank goodness, more consistency across Medline platforms when the search queries are complex (for example, during multi-set systematic review searching)? If so, that would be worth emphasizing or at least surmising yes or no in the conclusion for the simple reason that it will reassure systematic searchers.

We are honestly not sure. We seriously think more research needs to be done. This is because one complex search is not like the other. For example, does complex mean that a query simply mixes search fields, includes one or more Boolean operator, includes a mix of controlled terms and string-based searches? We do not know. We think the importance of this paper is that it raises doubts about this and to, hopefully, provide some justification for more research. We added some text to help clarify this.

I read a similar paper you authored published last year in the Journal of the Medical Library Association: Burns CS, Shapiro RM, II TN, Huber JT. Search results outliers among MEDLINE platforms. Journal of the Medical Library Association: JMLA. 2019 Jul;107(3):364.

- In comparing the two papers, apart from testing the searches over time, how much of the concept and analysis here are expanded upon and/or different from your 2019 paper? You do allude to similar work you have performed in the past early in your manuscript but you don't discuss this paper specifically unless I missed it. The titles of the two papers are different; are they, in your view, sufficiently different / distinct both in terms of conceptual design and methods to warrant two separate manuscripts? If they are connected, by virtue of concept or overall goals for your research, you should say so.

- Further, the JMLA paper in 2019 is well-written though the PLoS manuscript is more detailed and of course deals with the longitudinal aspects. The title of the 2019 manuscript describes the paper accurately; it would be worth adapting that title to your PLoS manuscript, using the same economy of words and approach, to connect the two papers. The current title does not in my view convey the subject content of the manuscript in a descriptive way.

We agree and thank you for the suggestion and for reading the prior paper. We have added additional text in the methods and conclusion sections to build off that JMLA paper and have changed the title of this paper to better reflect the link between the two.

- In comparing your two efforts, I'm aware of added sections in the current manuscript pertaining to calendar year (2020) with regard to searches for COVID-19. That is timely and potentially useful but the question is still whether this adds substantively to our understanding.

It shows that the MEDLINE platforms are not updated equally and at different rates. Again, we think that if this is known before, it’s only known anecdotally, and it’s important to document this scientifically. We think that’s very important for understanding big differences between these platforms and why it might be especially important to cite the specific MEDLINE platform used in studies that depend upon them for data. We have made that statement clearer in our conclusion.

On line 396, you indicate that Medline "contains more than 25 million references to journal articles" but this number is by now in 2020 considerably higher than that, and possibly now as high as 28 million.

- For this review, I have had the privilege of being able to read the previous peer reviewers' comments of your paper, and your responses.

- I agree with the previous peer reviewers' comments generally but in particular want to emphatically agree with the comments / question pertaining to how you devised the search queries. It's unclear to me, even after reading the methods section again, why you chose to focus on the queries you did as a basis for testing consistency. Did the medical librarian on the team provide insight? I keep going back to comments earlier in this review in that I would have thought truly complex search queries would be more revealing.

We revised the text somewhat but we’re not sure how we can be clearer. The queries were designed to be more simple than complex and to include basic field searches. We focused on straightforward searches in order to be able to track any problems we might find, and this is something we would not be able to do if were to add many terms and fields to our queries. For example, if I do a title name search in ProQuest/MEDLINE and one in EBSCOHost/MEDLINE, then any differences in search results can be explained by how these vendors index title names. But if I append to that query several other fields, then I no longer have a very good way to understand the differences between the two. 

- The focus on quantity of search results / retrieval numbers over any substantive discussion of quality of those results ("what exactly did you find when the numbers didn't agree across platforms? why did those extra references / citations appear in the retrieval sets?") is a major deficiency in the paper (which you acknowledge in your limitations section).

- In an ideal manuscript, it would be advisable to have a balance of both quantitative and qualitative aspects of your query analysis; perhaps your research team could have waited to publish until that part of your work was completed as you mention this paper is meant to be the start of a larger evaluation process.

We disagree. Additional research will be built up on this, but we think this stands on its own and will function as a building block for additional, qualitative research projects.

- What do I tell my students and other expert searchers/ users who might read this paper? I hope you are not suggesting we search multiple Medline platforms to compare results ourselves for simple search queries, or to look for additional papers that the individual platforms were unable to find.

We think that you tell them that MEDLINE is not MEDLINE is not MEDLINE. That the differences between platforms is multidimensional and, like you stated earlier, probably different from one year to the next as these databases are updated and as the technology changes and evolves. We think that you tell them that that if they search EBSCOhost/MEDLINE, then they should not expect the same results as someone who searches ProQuest/MEDLINE, and so forth, and that this might raise problems. We also think that you tell them that if they publish or work with others, they should clearly document which platform that they use. We added some text to help clarify this.

- Finally, as a medical librarian and information retrieval expert, what does this paper tell me and how does it inform my work? In the final analysis, it is not clear to me what your message is - other than to watch for variable results in simple searches on the various platforms. Perhaps you should send your manuscript to all the vendors for comment and so they can fix the problems.

Hopefully they read this. It’s not just vendors that need to know this. Users need to know, too. And it’s also very much a holistic issue, such that the problem is not just with vendors, but with trends in scholarly publication altoghether. So others need to be aware of this, too. Ovid did reach out to us after the JMLA paper was published and offered us free access for one year to their version of PubMed/MEDLINE. We declare this in the conflict of interests, and they had no input in the study. But it shows that they are interested in this kind of literature.

- Best of luck.

Thanks!

---

## [Decision Letter · Decision Letter 2]

7 Dec 2020

PONE-D-20-15022R2

MEDLINE search retrieval issues: A longitudinal query analysis across select vendor platforms

PLOS ONE

Dear Dr. Burns,

Thank you for submitting your manuscript to PLOS ONE. After careful consideration, we feel that it has merit but does not fully meet PLOS ONE’s publication criteria as it currently stands. Therefore, we invite you to submit a revised version of the manuscript that addresses the points raised during the review process.

As you can see from the reviewers' comments at the end of this email, they agree the manuscript in in much improved shape now. However, they have asked for some revisions that mainly relate to the language of the manuscript and how the results are presented.

We look forward to receiving your revised manuscript.

Kind regards,

Muhammad A. Z. Mughal, PhD

Academic Editor

PLOS ONE

Reviewers' comments:

Reviewer's Responses to Questions

**Comments to the Author**

1. If the authors have adequately addressed your comments raised in a previous round of review and you feel that this manuscript is now acceptable for publication, you may indicate that here to bypass the “Comments to the Author” section, enter your conflict of interest statement in the “Confidential to Editor” section, and submit your "Accept" recommendation.

Reviewer #1: (No Response)

Reviewer #3: (No Response)

2. Is the manuscript technically sound, and do the data support the conclusions?

Reviewer #1: Yes

Reviewer #3: Partly

3. Has the statistical analysis been performed appropriately and rigorously? 

Reviewer #1: Yes

Reviewer #3: I Don't Know

4. Have the authors made all data underlying the findings in their manuscript fully available?

Reviewer #1: Yes

Reviewer #3: Yes

5. Is the manuscript presented in an intelligible fashion and written in standard English?

Reviewer #1: Yes

Reviewer #3: Yes

6. Review Comments to the Author

Reviewer #1: PLOS1 review

Please note that line numbers refer to the version that includes markup.

This paper is much better but there are still issues:

* the terms “database”, “index”, and “platform” are conflated. This makes it very hard for a non-expert (someone who does not perform medical searching as an expert) to follow the argument. Examples of conflations that must be fixed are below. It would help tremendously if you started out by defining these three terms and then using them consistently and rigorously throughout.

* As one of your reviewers pointed out, library experts already know all this – in detail. I keep suggesting that your paper be revised rather than rejected because I believe it is important that non-experts also understand your findings. You are still not writing for that non-expert audience. My suggestion is that you write a paragraph in the introduction to make the following delineations:

Medline bibliographic records

-------- comprise -------

the Medline database

------- which is -------

indexed in different ways

------- by ----------

platforms

------- used for search and access for Medline records -------------

The point is that access and search over what should be a stable set of Medline records is unreliable and inconsistent depending on anecdotal factors your study has quantified. It might help to think about these issues as if you were a computer programmer who is familiar with the concept of database records, indexing, and information retrieval over structured records. While there are nuances in the way librarians think about and talk about these distinctions, it is often very hard to follow a discussion that conflates these things.

* You introduce “MESH” on line 64 but do not define or explain it. Drop the reference to LCSH – it is unimportant to your point. Non-expert readers will not make the connection between these two things. Use the space to explain what MESH is and why it matters. Below I also point out places where understanding the MESH concept is critical.

*Similarly, in the next two paragraphs you refer to ERIC, DOE, NLM, etc.. Stick to the point you are making without drawing parallels – for your non-expert reader these won’t be meaningful. They just get in the way of understanding your point. Your reader needs to understand the major parts (records, database, index, platform) in order to understand how the system breaks down on a longitudinal basis. By the end of the intro these components should be clear to someone who has never heard of Medline.

* line 75: don’t say “presumably” – your reader is counting on you to say something does, may, or does not pertain; also, “access points” is librarian talk --- these are index fields

* line 132: don’t say “It seems that” – same point, your reader is counting on you to say something does, may, or does not pertain. Is it true that “many health science informational professionals and librarians are aware of the

133 issues that we report here”? – then just say that.

* Much of the material in paragraphs between line 67 and line 81 can be simplified and combined with the clear paragraph that runs from 82 to 89. As suggested above, the parallels are not helpful to your explanation.

* You have presented PubMed as a platform in most of the paper but in the introduction you imply that it is an intermediate step between the Medline index and the proprietary platforms. The role of PubMed is muddled. I have thought of two options – and I like my second idea best.

1) Maybe you can refer to the PubMed WEBSITE as the platform and simply indicate that MESH terms are added to the Medline index before it is distributed to vendors. The fact that the MESH-enhanced index “base” is called PubMed just adds needless confusion about details that are less important (paragraph between lines 82 and 89). This is related to my point on the figures – the nuances are confusing, don’t help your reader understand, and detract from the important findings.

OR

2) Justify using the MESH indexed PubMed as the standard against which other platforms are measured. I think this is a better approach. In this case you need to explain that the MESH-enhanced PubMed index is the foundation used by the other platforms. This makes sense, but you need to get into some version of this:

Medline bibliographic records

-------- comprise -------

the Medline database

------- which is indexed using MESH terms to create

the PubMed database

------- which is further enhanced with additional index fields by the ----------

platforms (including PubMed WEBSITE)

------- that make Medline records accessible for searching -------------

In any event, you are referring to PubMed using two difference constructs and it is confusing. Rather then try to explain the confusion, just use language that clearly separate the two rolls of “PubMed”.

* These sentences are very unclear (line 103 – 106): “If searchers used too few databases to conduct literature searches [32–34], then they may miss relevant studies [10,35]. … This is especially important in cases where data from past research is collected, analyzed, and synthesized based on published and/or gray literature,…” – Is this referring to too few platforms? Or are you saying that searchers must use Medline, additional databases, gray literature? These passages are not helping your reader. They go off the track of thinking about how variability in indexes and platforms affects query performance regarding longitudinal consistency. The point of this paragraph is very muddled.

* At line 134 you state “We are not aware of what the broader bio-science and medical communities know because few studies have included queries that were designed to test reproducibility be reproducible across systems.” The sentence does not make sense? You’re not aware because of the lack of studies? The revised paragraph is muddled. One issue is that non-experts have little or no information. The other is that prior studies have not been done to measure and confirm the anecdotal knowledge/reports. Both points are important and both are lost in the first part of the paragraph.

* You must explain the concepts of “MESH branch” and “exploded MESH” in language that a database designer, information science person, or computer scientist would use. Understanding your method and results depends on understanding of these concepts. Relate the ideas to the more general notions of index terms, tags, and tree structure.

* The methods section is much easier to follow. Tweaks:

** line 182 “(e.g., all queries including a single keyword search in a journal title field in a case)” is confusing and not needed

** line 186 “were designed to search all five MEDLINE platforms” - to search EACH of the MEDLINE platforms

** line 189 – that the data WERE collected

** lines 193 – 202 – my version: We designed our 29 cases using basic search fields to examine compare search retrieval counts on each of the five MEDLINE platforms, therefore, our queries were designed to be short, logically clear, and to enhance the comparability of the results. For example, Table 1 describes a simple MeSH search, limited by publication date range.

** in the lines that follow you report your results as if this is a justification for the method. You can state this more convincingly with something like this:

Our prior findings [cite yourself if this is true – our cite someone else – or simply say “based on our preliminary testing”] indicated that a wide number of records would be retrieved by different platforms, particularly when MESH terms were added to the strategies/queries. On this basis, we created 29 cases then to test a broad range of basic search strategies.

** line 235 – “We used the modified z-score to locate deviations around results from the PubMed MEDLINE platform” – this must be justified. Why is PubMed the logical standard? (see above). Explaining this will go a long way toward making the results easier to understand. This is about the indexes. Somewhere be sure you are clearer that indexing and query structure go hand in glove. This helps explain index differences as a source of critical variability in results. Upload timing and errors are easy to understand. Indexing and retrieval differences are not.

Conflations of “database”, “index” and “platform”:

* In your abstract, 22-23 “Specific bibliographic databases, like PubMed and MEDLINE….” – but you’ve set this up with MEDLINE as the database – and you present PubMed as one of several platforms. Conflating them in the abstract is very confusing.

* line 31 “Bibliographic databases are used to identify and collect research data… ” – here you are talking about platforms, not databases

* line 33 “These systems, in particular, PubMed and MEDLINE, are used to inform clinical….” – just say “these systems” to avoid confusion about the differences between PM and ML. Stick with the simple, clear and correct but incomplete definition of Medline as a core underlying database of bibliographic records and PubMed as a platform that has its own index of the Medline records

* line 36 “…the development of bibliographic databases or…” – the development of bibliographic platforms

* line 58 “…bibliographic databases, such as those available on…” – you are talking about indexes here

* lines 62-63 “In some specialized databases, these records may be supported” – you are talking about indexes here

* line 90 “This differentiation among database platforms has” – this is differentiation between platforms

* line 97 “the same MEDLINE data file” – same MEDLINE database

* line 101 “Although the choice of database systems impacts potential source” – this is the choice of platforms

* line 102 “it is not known how searching the sdame database (e.g., MEDLINE) on different platforms” – simplify this === “it is not known how searching .. the MEDLINE database… on different platforms … ” – that’s what you are talking about

* line 117 “principles are applied by database vendors in indexing” -- these are platform vendors – or just platforms

* line 120 “problematic in bibliographic databases purchased by libraries” – bibliographic platforms – ore more generally “bibliographic resources” – reserve “database” to refer to Medline only

* line 125 “license these records to database vendors to host on their own platforms” – to vendors – you don’t need to call them database vendors

Finally, your results section is still extremely unclear and profusion of charts is distracting and confusing. All this seriously undermines your point. The notion of macro and micro are also not useful. Rather then giving these two perspectives unnecessary and confusing names, just write what what you are talking about – what you did. The idea of analysis “scale” is also not useful. In order to understand what you are reporting I need to revert to looking at your raw data. In short, major changes are still needed to make the results meaningful for non-expert readers. I believe only a determined expert will be motivated to decipher it.

In general, Figure 3 is meaningless. You can show one example if you wish, but the text should convey that whatever it is you are measuring in those charts is a constant month over month. The results displayed in Figures 4, 5 and 6 are meaningful, but the meaning is buried in the complexity of the text and in all the charts. In my opinion, your results section should focus on making very clear the meaning and implications of data displayed in figs. 4, 5, 6, but you need to select only one chart as an example for each type of anomaly you have found. You can make all the other charts available online with the data. The details in figures 7 and 8 are also obscure your point. You can report that information much more effectively with one example figure and a summary table on your stats.

Many of the details of your results section are dense and hard to understand, particularly without definitions of things like MESH and “explode”. If there are important esoteric nuances to the specific cases, that detail may be better suited for publication in a second paper written specifically for experts.

With a greatly simplified presentation of your results readers will be able to follow. This will make your final chart (Fig 9) much more meaningful. This is a nice, clear and current demonstration of the effect of the issues you’ve characterized.

Reviewer #3: This is my second review of your paper. You and your team have made progress in some areas of the paper; some parts of the paper are coming together, but other parts are still unclear.

I've made extensive editorial comments (N=34 in total) focussing on using simpler, more consistent language in the manuscript. Some sections in the paper are still difficult to understand; I've done my best to try to assist in clarifying your points but not certain I've achieved much. Best of luck with your research.

My comments are:

1) I like the new title for the article, "MEDLINE search retrieval issues: A longitudinal query analysis across select vendor platforms" though I would take out the word 'select' and replace it with the word, "five".

2) Abstract: Your first sentence is too long and somewhat confusing. Can you consider a change to the second sentence at line 14 as, "We devised twenty-nine search queries, or cases, comprised of five semantically equivalent queries per case to search against five MEDLINE platforms." Lines 17-23 are important and I'd recommend breaking up the section by saying, "We found that search results varied by MEDLINE platform within sets and across time. Reasons for variations were due to trends in scholarly publishing such as publishing individual papers online first versus complete issues in print format. Some other reasons were metadata differences in bibliographic records; differences in levels of specificity of search fields across platforms and large fluctuations in monthly search results based on the same query. Database integrity and currency issues were observed as each platform updated its MEDLINE files throughout the year."

3) Line 35 should say "...in the health professions and to construct a knowledge base for clinical decision support systems".

4) Line 49 remove "should".

5) Line 53 sounds a bit odd with "relies"; I would use rely for better agreement.

6) Lines 54-58 could be shortened as "The Preferred Reporting Items for Systematic Reviews and Meta-Analyses (PRISMA) Guidelines and the Cochrane Handbook for Systematic Reviews of Interventions provide examples for scholars in their reporting of methods and organizing of reviews".

7) At line 60, it's not that they rely on structured records but that they are designed that way for descriptive purposes and to create reliable post-hoc searchable citations/ records.

8) This sentence is unnecessarily awkward at line 61, "These bibliographic records contain fields we take as meaningful for providing discovery and access, such as author name fields, document title fields, publication title fields, and date of publication fields." Perhaps take out "we take as meaningful'.

9) At line 65, I would not use this phrase, "are based on standard knowledge classification efforts". I would not refer to subject analysis and/or article indexing as knowledge classification for obvious reasons.

10) Line 75, I don't really understand why you use the phrase "Commercial information service companies" when you can say simply "Commercial vendors such as" or simply "vendors". I wouldn't end the sentence at line 78 with a preposition. I would say "database providers, such as the NLM or DOE, etc" and remove the word 'original' which is not accurate in any case.

11) Line 78, isn't it really about database "features" and not "value"? What do you mean by value? Not sure I agree that it's about value and not about searchable features.

12) Commercial information service provider at line 86, say "vendor". You don't need to keep changing the phrase which will only confuse your reader. (You also continue this practice throughout the manuscript).

13) Line 90 why say "differentiation"? when you can say "Differences in features across platforms"? And I'm not sure that accounting for those differences is why we report the specific platforms, interfaces and / or vendors. It's to help reproduce the searches as reported by using the correct platform.

14) Line 97 I would remove "presumably". It's a kind of cynical word to use in this context.

15) Line 101: this sentence seems to relate more to the previous sentence than what comes after it.

16) Line 114: "database systems" should be vendor or "vendor systems" or "database vendors". If clarity is needed, you can say "Vendor systems / or vendors such as OVID or EBSCOHost".

17) Line 121: I'd remove "also". There are too many examples of unneeded uses of this word in your paper; many are not needed.

18) Line 132: Remove "It seems that" and the sentence needs a citation. I would argue that varying results by vendor platform have been seen in other papers and you could cite one or two. Cite your previous paper from JMLA for example.

19) Line 144: Change this sentence to something along the lines of "This paper builds on that work and examines differences over time by evaluating longitudinal data, which is a critical factor in replicating bibliographic database search results. "

20) Line 162: This first sentence is terribly confusing to me. I would say, "We designed 29 cases, or sets, of searches which comprised five (5) semantically similar or equivalent queries to perform searches in five MEDLINE platforms for a total of 145 searches".

21) Lines 180-188: can you say here why you picked dementia? Because it's one word? Because it is short and easy? Some rationale here is really needed even if you feel you are repeating information. It will help your reader understand why you picked it and grouped it with neoplasia.

22) Line 247: Can you set up this section for clarity? For example, "The data gathered in the majority of our searches (20 cases of five searches each for a total of 100 queries) did not include limits on publication dates. However, we did include date delimited searches for nine cases of five searches for a total of 45 queries. We used publication date limits from 1950 to 2015 for these search queries. We found that...etc."

23) Line 259: This sentence is awkward. Can you say simply, "Second, based on a detailed examination of the same cases...". Remove the use of the word also (2x) from lines 263 and 268.

24) Line 272: This whole section on macro/micro is more confusing than it needs to be. I would consider using more simple language. For example "We examined the data of the cases, and present variations we identified based on two categories which we will use in the following section: an overview (macro) of the data and a more granular (or, micro) view of the variations arising in the data".

25) Line 323: I would relabel this section as "Online first and print publications impact bibliographic control". Also line 327: I would avoid the phrase "non exploding". Can you say "...not exploded, but differed because case #04 is restricted by publication date". Please use simpler sentence construction for the benefit of communicating your results. Throughout this section, I would look critically at using the phrase "non exploding" as this is not the way this is referred to in describing the feature. Say "not exploded".

26) Line 339: I would not say "indicates"; perhaps "suggests" would make more sense.

27) Line 348: Change along these lines, "The record indicates it was added to PubMed in 2015 but not formally entered into MEDLINE until 2019".

28) Lines 361-378 should be edited for brevity. It's also very acronym heavy.

29) Line 379: what do you mean by the title of this section "Reproducible with specific field searches"? Do you mean "Reproducibility improved with search specificity (or field searching)"

30) Lines 394-397: the label for this section is a bit unclear as is the first sentence of the section. Perhaps it's the use of the word "outliers". What does this mean? Just different from the rest? An error? An incorrectly inflated number?

31) Lines 400-522: this section is quite difficult to read. Without going into detail, I'd recommend a close edit. For example, at Line 488-490, try something simpler such as "We could not identify the reasons for the significant variance in search results over time. As such, searchers may be left wondering why they are seeing variances in search results which may further undermine their trust in using the different platforms".

32) In your Limitations section, you need an introductory sentence. "Our paper has a number of limitations based on X, Y and Z." As it's written in the first sentence, I don't understand what you mean by saying that using PubMed / Medline is a limitation (due to different coverage and content or...?).

33) Line 523: why is documenting variances in systematic searches across platforms a useful area of research? Could you explain this to the reader? It might be important - but your explanation doesn't really clarify. Perhaps you could say that documenting variances on platforms is NOT as important as trying to identify which vendor platform is the most reliable over time, and why. That's the issue we as searchers want to know about and why we are reading your paper. Perhaps a ranked list would be a good research project. It shouldn't be up to health librarians or expert searchers to document these variances. They take up our time! to do searches. I think the semantically selected search queries are a limitation also but you don't really know because you haven't tested them. You might say this may have affected generalizability.

34) Conclusions in your paper: Not sure why you introduce Google Scholar in your lead sentence of your conclusion. I'd reconsider that first sentence very closely. Also, what's the most important concluding remark about your research after all the work you put into it? Perhaps it's that "No single MEDLINE platform emerged as the most reliable interface to search for citations for this project". The data suggests that there were greater variances in search results over time for simpler, shorter search queries and that the more sophisticated search queries performed more reliably over the five vendor platforms. Whatever it is, can you be explicit? Best of luck.

/dg

7. PLOS authors have the option to publish the peer review history of their article (what does this mean?). If published, this will include your full peer review and any attached files.

Reviewer #1: No

Reviewer #3: **Yes: **Dean Giustini

---

## [Author Response · Author response to Decision Letter 2]

8 Jan 2021

Reviewer #1: PLOS1 review

Please note that line numbers refer to the version that includes markup.

This paper is much better but there are still issues:

* the terms “database”, “index”, and “platform” are conflated. This makes it very hard for a non-expert (someone who does not perform medical searching as an expert) to follow the argument. Examples of conflations that must be fixed are below. It would help tremendously if you started out by defining these three terms and then using them consistently and rigorously throughout.

We have made some effort to clarify the differences in the introduction by describing how these systems function. We think this is clear. We also would like to push back a bit. PLOS publishes thousands of articles each year in a number of different subject areas. Many of these published articles discuss specialized areas of research. While we think it’s important to write clearly, and we thank you for helping us clarify some of our language, we are not writing for the general public, just as the thousands of articles that are published PLOS are not written for the general public. We think that health science librarians and informationists and others in the broader bioscience fields who uses these systems will understand this article.

* As one of your reviewers pointed out, library experts already know all this – in detail. I keep suggesting that your paper be revised rather than rejected because I believe it is important that non-experts also understand your findings. You are still not writing for that non-expert audience.

We think this is an editorial position and that it does not affect the soundness of this research. Therefore, respectfully, we think this opinion is out of bounds.

Our goal is to write and communicate this research for the “library expert” audience, but we are also writing for the medical, bioscience, and like audiences, too. These audiences should already be familiar with MeSH, PubMed, MEDLINE, etc., as it is a regular part of medical training. Beyond that training, medical doctors and like also conduct systematic reviews, and are familiar with MeSH, etc already. And so we strongly contest the argument that this should be written for a non-expert audience or that the non-expert audience is limited to who you may be imagining. And with all due respect, we reiterate that we do not think it is fair that this paper is being judged this way. There are thousands of papers on PLOS ONE that are highly technical and scientific. We’re not sure, therefore, what is motivating this need to appeal to a non-expert audience or why you think that the expert audience is rather limited. Also, even though one of the reviewers pointed out that “library experts already know all this”, we have also shown, and rebutted in our last review, that they only know this anecdotally. Anecdotal knowledge is not scientific knowledge. This should be better evidence than anecdotal. And even if it is known among “library experts”, many of these experts still are not clear in their papers which version of MEDLINE that they are using, as we have discussed in the paper, which is why we make it clear that they should do that in their papers when we address this in our discussion and conclusion. This misunderstanding or lack of knowledge about the problems with MEDLINE also are not necessarily known among other users of MEDLINE, etc., who are experts in some ways with it or do know the basics, but who may not know the problems that we are addressing here.

My suggestion is that you write a paragraph in the introduction to make the following delineations:

Medline bibliographic records

-------- comprise -------

the Medline database

------- which is -------

indexed in different ways

------- by ----------

platforms

------- used for search and access for Medline records -------------

Thank you for your suggestion, but we believe it is clear as it stands.

The point is that access and search over what should be a stable set of Medline records is unreliable and inconsistent depending on anecdotal factors your study has quantified. It might help to think about these issues as if you were a computer programmer who is familiar with the concept of database records, indexing, and information retrieval over structured records. While there are nuances in the way librarians think about and talk about these distinctions, it is often very hard to follow a discussion that conflates these things.

We believe using computer programmer language would only confuse the issue. It’s already difficult to use terms like “database”, which has multiple meanings. We have made every attempt to be clear in our language and we believe we should not be faulted for how people with different field backgrounds might read this.

* You introduce “MESH” on line 64 but do not define or explain it. Drop the reference to LCSH – it is unimportant to your point. Non-expert readers will not make the connection between these two things. Use the space to explain what MESH is and why it matters. Below I also point out places where understanding the MESH concept is critical.

We have dropped the reference to LCSH in this sentence and have added an extra clause that further define MeSH.

* Similarly, in the next two paragraphs you refer to ERIC, DOE, NLM, etc.. Stick to the point you are making without drawing parallels – for your non-expert reader these won’t be meaningful. They just get in the way of understanding your point. Your reader needs to understand the major parts (records, database, index, platform) in order to understand how the system breaks down on a longitudinal basis. By the end of the intro these components should be clear to someone who has never heard of Medline.

We have dropped the references to these parallel systems.

* line 75: don’t say “presumably” – your reader is counting on you to say something does, may, or does not pertain; also, “access points” is librarian talk --- these are index fields

We used the term “access” only and have removed points.

* line 132: don’t say “It seems that” – same point, your reader is counting on you to say something does, may, or does not pertain. Is it true that “many health science informational professionals and librarians are aware of the

133 issues that we report here”? – then just say that.

Revised.

* Much of the material in paragraphs between line 67 and line 81 can be simplified and combined with the clear paragraph that runs from 82 to 89. As suggested above, the parallels are not helpful to your explanation.

Revised and combined.

* You have presented PubMed as a platform in most of the paper but in the introduction you imply that it is an intermediate step between the Medline index and the proprietary platforms. The role of PubMed is muddled. I have thought of two options – and I like my second idea best.

 1) Maybe you can refer to the PubMed WEBSITE as the platform and simply indicate that MESH terms are added to the Medline index before it is distributed to vendors. The fact that the MESH-enhanced index “base” is called PubMed just adds needless confusion about details that are less important (paragraph between lines 82 and 89). This is related to my point on the figures – the nuances are confusing, don’t help your reader understand, and detract from the important findings.

OR

2) Justify using the MESH indexed PubMed as the standard against which other platforms are measured. I think this is a better approach. In this case you need to explain that the MESH-enhanced PubMed index is the foundation used by the other platforms. This makes sense, but you need to get into some version of this:

Medline bibliographic records

-------- comprise -------

the Medline database

------- which is indexed using MESH terms to create

the PubMed database

------- which is further enhanced with additional index fields by the ----------

platforms (including PubMed WEBSITE)

------- that make Medline records accessible for searching -------------

In any event, you are referring to PubMed using two difference constructs and it is confusing. Rather then try to explain the confusion, just use language that clearly separate the two rolls of “PubMed”.

We think our discussion of PubMed is clear as it stands.

* These sentences are very unclear (line 103 – 106): “If searchers used too few databases to conduct literature searches [32–34], then they may miss relevant studies [10,35]. … This is especially important in cases where data from past research is collected, analyzed, and synthesized based on published and/or gray literature,…” – Is this referring to too few platforms? Or are you saying that searchers must use Medline, additional databases, gray literature? These passages are not helping your reader. They go off the track of thinking about how variability in indexes and platforms affects query performance regarding longitudinal consistency. The point of this paragraph is very muddled.

We have divided this into two separate paragraphs and have revised to make it clearer.

* At line 134 you state “We are not aware of what the broader bio-science and medical communities know because few studies have included queries that were designed to test reproducibility be reproducible across systems.” The sentence does not make sense? You’re not aware because of the lack of studies? The revised paragraph is muddled. One issue is that non-experts have little or no information. The other is that prior studies have not been done to measure and confirm the anecdotal knowledge/reports. Both points are important and both are lost in the first part of the paragraph.

We inserted the anecdotal sentence here based on the other reviewer’s comments in the prior review and that you reference above, but this is also anecdotal – that is, whether such library experts are even anecdotally aware of these issues, or if many are, how many or how few are. In many ways, these peer reviews have been very helpful, but they are also causing some of what we are saying to get muddled. So we have removed the anecdotal sentence here because we think it’s caused some of the confusion that you point out here. Besides, we are not sure how well this is anecdotally known, even though the other reviewer claims that it is well known.

* You must explain the concepts of “MESH branch” and “exploded MESH” in language that a database designer, information science person, or computer scientist would use. Understanding your method and results depends on understanding of these concepts. Relate the ideas to the more general notions of index terms, tags, and tree structure.

* The methods section is much easier to follow. Tweaks:

** line 182 “(e.g., all queries including a single keyword search in a journal title field in a case)” is confusing and not needed

Removed.

** line 186 “were designed to search all five MEDLINE platforms” - to search EACH of the MEDLINE platforms

Revised.

** line 189 – that the data WERE collected

Revised.

** lines 193 – 202 – my version: We designed our 29 cases using basic search fields to examine compare search retrieval counts on each of the five MEDLINE platforms, therefore, our queries were designed to be short, logically clear, and to enhance the comparability of the results. For example, Table 1 describes a simple MeSH search, limited by publication date range.

Revised.

** in the lines that follow you report your results as if this is a justification for the method. You can state this more convincingly with something like this:

Our prior findings [cite yourself if this is true – our cite someone else – or simply say “based on our preliminary testing”] indicated that a wide number of records would be retrieved by different platforms, particularly when MESH terms were added to the strategies/queries. On this basis, we created 29 cases then to test a broad range of basic search strategies.

Revised.

** line 235 – “We used the modified z-score to locate deviations around results from the PubMed MEDLINE platform” – this must be justified. Why is PubMed the logical standard? (see above). Explaining this will go a long way toward making the results easier to understand. This is about the indexes. Somewhere be sure you are clearer that indexing and query structure go hand in glove. This helps explain index differences as a source of critical variability in results. Upload timing and errors are easy to understand. Indexing and retrieval differences are not.

Revised.

Conflations of “database”, “index” and “platform”:

* In your abstract, 22-23 “Specific bibliographic databases, like PubMed and MEDLINE….” – but you’ve set this up with MEDLINE as the database – and you present PubMed as one of several platforms. Conflating them in the abstract is very confusing.

We do not think we are not conflating these terms. In the first sentence, this is clear with “… of the MEDLINE database hosted on multiple platforms”.

* line 31 “Bibliographic databases are used to identify and collect research data… ” – here you are talking about platforms, not databases

“Bibliographic databases” is a legitimate term for this. Here we are not referring to any platform but to bibliographic databases generally.

* line 33 “These systems, in particular, PubMed and MEDLINE, are used to inform clinical….” – just say “these systems” to avoid confusion about the differences between PM and ML. Stick with the simple, clear and correct but incomplete definition of Medline as a core underlying database of bibliographic records and PubMed as a platform that has its own index of the Medline records

Removed “These systems”, but PubMed is more than MEDLINE and includes records not indexed in MEDLINE.

* line 36 “…the development of bibliographic databases or…” – the development of bibliographic platforms

This is not necessary to change. Platforms are much more than the bibliographic databases. Here we are referring to databases.

* line 58 “…bibliographic databases, such as those available on…” – you are talking about indexes here

We are talking about databases that exist on the platforms. We lightly revised.

* lines 62-63 “In some specialized databases, these records may be supported” – you are talking about indexes here

No, a database creates an index. We’re talking about databases. Our language is correct.

* line 90 “This differentiation among database platforms has” – this is differentiation between platforms

Revised.

* line 97 “the same MEDLINE data file” – same MEDLINE database

Revised.

* line 101 “Although the choice of database systems impacts potential source” – this is the choice of platforms

Revised.

* line 102 “it is not known how searching the sdame database (e.g., MEDLINE) on different platforms” – simplify this === “it is not known how searching .. the MEDLINE database… on different platforms … ” – that’s what you are talking about

Revised.

* line 117 “principles are applied by database vendors in indexing” -- these are platform vendors – or just platforms

Revised.

* line 120 “problematic in bibliographic databases purchased by libraries” – bibliographic platforms – ore more generally “bibliographic resources” – reserve “database” to refer to Medline only

Revised.

* line 125 “license these records to database vendors to host on their own platforms” – to vendors – you don’t need to call them database vendors

Revised.

Finally, your results section is still extremely unclear and profusion of charts is distracting and confusing. All this seriously undermines your point. The notion of macro and micro are also not useful. Rather then giving these two perspectives unnecessary and confusing names, just write what what you are talking about – what you did. The idea of analysis “scale” is also not useful. In order to understand what you are reporting I need to revert to looking at your raw data. In short, major changes are still needed to make the results meaningful for non-expert readers. I believe only a determined expert will be motivated to decipher it.

We disagree that this is confusing and that the terms we’re using are not useful. We already know that our results have been looked at based on the comments via social media about our preprint (https://www.biorxiv.org/content/10.1101/2020.05.22.110403v1) and that our points have been understood. Figure 3 is an important contrast to Figures 4, 5, and 6, and the terms “micro” and “macro” have no special meaning beyond their everyday definitions. However, based on the other reviewer’s comments, we have clarified the language and research involving the macro and micro views.

In general, Figure 3 is meaningless. You can show one example if you wish, but the text should convey that whatever it is you are measuring in those charts is a constant month over month. The results displayed in Figures 4, 5 and 6 are meaningful, but the meaning is buried in the complexity of the text and in all the charts. In my opinion, your results section should focus on making very clear the meaning and implications of data displayed in figs. 4, 5, 6, but you need to select only one chart as an example for each type of anomaly you have found. You can make all the other charts available online with the data. The details in figures 7 and 8 are also obscure your point. You can report that information much more effectively with one example figure and a summary table on your stats.

We strongly disagree that these figures are meaningless or that they obscure our point. If you feel you need to recommend rejecting this article for these reasons, I guess go ahead. We’ve already defended these plots and we think they are important visualizations. 

Many of the details of your results section are dense and hard to understand, particularly without definitions of things like MESH and “explode”. If there are important esoteric nuances to the specific cases, that detail may be better suited for publication in a second paper written specifically for experts.

Again, we think this criticism is unfair and does not impact the soundness of our research. We are not writing for the general public, just as most articles on PLOS are not for the general public. We appreciate your recommendations above and previously. You have helped us clarify the language and the terms, which are confusing for a number of reasons beyond this research. But we can only go so far in making this paper accessible to the “non-expert”. 

With a greatly simplified presentation of your results readers will be able to follow. This will make your final chart (Fig 9) much more meaningful. This is a nice, clear and current demonstration of the effect of the issues you’ve characterized.

Reviewer #3: This is my second review of your paper. You and your team have made progress in some areas of the paper; some parts of the paper are coming together, but other parts are still unclear.

I've made extensive editorial comments (N=34 in total) focusing on using simpler, more consistent language in the manuscript. Some sections in the paper are still difficult to understand; I've done my best to try to assist in clarifying your points but not certain I've achieved much. Best of luck with your research.

My comments are:

1) I like the new title for the article, "MEDLINE search retrieval issues: A longitudinal query analysis across select vendor platforms" though I would take out the word 'select' and replace it with the word, "five".

Revised.

2) Abstract: Your first sentence is too long and somewhat confusing. Can you consider a change to the second sentence at line 14 as, "We devised twenty-nine search queries, or cases, comprised of five semantically equivalent queries per case to search against five MEDLINE platforms." Lines 17-23 are important and I'd recommend breaking up the section by saying, "We found that search results varied by MEDLINE platform within sets and across time. Reasons for variations were due to trends in scholarly publishing such as publishing individual papers online first versus complete issues in print format. Some other reasons were metadata differences in bibliographic records; differences in levels of specificity of search fields across platforms and large fluctuations in monthly search results based on the same query. Database integrity and currency issues were observed as each platform updated its MEDLINE files throughout the year."

Revised.

3) Line 35 should say "...in the health professions and to construct a knowledge base for clinical decision support systems".

Revised.

4) Line 49 remove "should".

Revised.

5) Line 53 sounds a bit odd with "relies"; I would use rely for better agreement.

Revised.

6) Lines 54-58 could be shortened as "The Preferred Reporting Items for Systematic Reviews and Meta-Analyses (PRISMA) Guidelines and the Cochrane Handbook for Systematic Reviews of Interventions provide examples for scholars in their reporting of methods and organizing of reviews".

7) At line 60, it's not that they rely on structured records but that they are designed that way for descriptive purposes and to create reliable post-hoc searchable citations/ records.

Revised.

8) This sentence is unnecessarily awkward at line 61, "These bibliographic records contain fields we take as meaningful for providing discovery and access, such as author name fields, document title fields, publication title fields, and date of publication fields." Perhaps take out "we take as meaningful'.

Revised.

9) At line 65, I would not use this phrase, "are based on standard knowledge classification efforts". I would not refer to subject analysis and/or article indexing as knowledge classification for obvious reasons.

Revised.

10) Line 75, I don't really understand why you use the phrase "Commercial information service companies" when you can say simply "Commercial vendors such as" or simply "vendors". I wouldn't end the sentence at line 78 with a preposition. I would say "database providers, such as the NLM or DOE, etc" and remove the word 'original' which is not accurate in any case.

Revised based on your comment and the other reviewer’s.

11) Line 78, isn't it really about database "features" and not "value"? What do you mean by value? Not sure I agree that it's about value and not about searchable features.

Just that adding features that are different than what can be found on PubMed is a selling point for them. But revised.

12) Commercial information service provider at line 86, say "vendor". You don't need to keep changing the phrase which will only confuse your reader. (You also continue this practice throughout the manuscript).

Revised throughout.

13) Line 90 why say "differentiation"? when you can say "Differences in features across platforms"? And I'm not sure that accounting for those differences is why we report the specific platforms, interfaces and / or vendors. It's to help reproduce the searches as reported by using the correct platform.

Revised.

14) Line 97 I would remove "presumably". It's a kind of cynical word to use in this context.

Revised.

15) Line 101: this sentence seems to relate more to the previous sentence than what comes after it.

Revised this section.

16) Line 114: "database systems" should be vendor or "vendor systems" or "database vendors". If clarity is needed, you can say "Vendor systems / or vendors such as OVID or EBSCOHost".

Revised.

17) Line 121: I'd remove "also". There are too many examples of unneeded uses of this word in your paper; many are not needed.

Revised.

18) Line 132: Remove "It seems that" and the sentence needs a citation. I would argue that varying results by vendor platform have been seen in other papers and you could cite one or two. Cite your previous paper from JMLA for example.

Revised based on this and other other reviewer’s comments.

19) Line 144: Change this sentence to something along the lines of "This paper builds on that work and examines differences over time by evaluating longitudinal data, which is a critical factor in replicating bibliographic database search results. "

Revised.

20) Line 162: This first sentence is terribly confusing to me. I would say, "We designed 29 cases, or sets, of searches which comprised five (5) semantically similar or equivalent queries to perform searches in five MEDLINE platforms for a total of 145 searches".

We have taken out the term sets because another reviewer didn’t like that term and suggested using cases. Otherwise revised.

21) Lines 180-188: can you say here why you picked dementia? Because it's one word? Because it is short and easy? Some rationale here is really needed even if you feel you are repeating information. It will help your reader understand why you picked it and grouped it with neoplasia.

Revised. Chosen just based on simplicity and for representation. The term was not grouped with neoplasms. Those were separate searches.

22) Line 247: Can you set up this section for clarity? For example, "The data gathered in the majority of our searches (20 cases of five searches each for a total of 100 queries) did not include limits on publication dates. However, we did include date delimited searches for nine cases of five searches for a total of 45 queries. We used publication date limits from 1950 to 2015 for these search queries. We found that...etc."

Revised.

23) Line 259: This sentence is awkward. Can you say simply, "Second, based on a detailed examination of the same cases...". Remove the use of the word also (2x) from lines 263 and 268.

Revised.

24) Line 272: This whole section on macro/micro is more confusing than it needs to be. I would consider using more simple language. For example "We examined the data of the cases, and present variations we identified based on two categories which we will use in the following section: an overview (macro) of the data and a more granular (or, micro) view of the variations arising in the data".

Thank you. Revised.

25) Line 323: I would relabel this section as "Online first and print publications impact bibliographic control". Also line 327: I would avoid the phrase "non exploding". Can you say "...not exploded, but differed because case #04 is restricted by publication date". Please use simpler sentence construction for the benefit of communicating your results. Throughout this section, I would look critically at using the phrase "non exploding" as this is not the way this is referred to in describing the feature. Say "not exploded".

Revised throughout.

26) Line 339: I would not say "indicates"; perhaps "suggests" would make more sense.Reviewer #1: PLOS1 review

Please note that line numbers refer to the version that includes markup.

This paper is much better but there are still issues:

* the terms “database”, “index”, and “platform” are conflated. This makes it very hard for a non-expert (someone who does not perform medical searching as an expert) to follow the argument. Examples of conflations that must be fixed are below. It would help tremendously if you started out by defining these three terms and then using them consistently and rigorously throughout.

We have made some effort to clarify the differences in the introduction by describing how these systems function. We think this is clear. We also would like to push back a bit. PLOS publishes thousands of articles each year in a number of different subject areas. Many of these published articles discuss specialized areas of research. While we think it’s important to write clearly, and we thank you for helping us clarify some of our language, we are not writing for the general public, just as the thousands of articles that are published PLOS are not written for the general public. We think that health science librarians and informationists and others in the broader bioscience fields who uses these systems will understand this article.

* As one of your reviewers pointed out, library experts already know all this – in detail. I keep suggesting that your paper be revised rather than rejected because I believe it is important that non-experts also understand your findings. You are still not writing for that non-expert audience.

We think this is an editorial position and that it does not affect the soundness of this research. Therefore, respectfully, we think this opinion is out of bounds.

Our goal is to write and communicate this research for the “library expert” audience, but we are also writing for the medical, bioscience, and like audiences, too. These audiences should already be familiar with MeSH, PubMed, MEDLINE, etc., as it is a regular part of medical training. Beyond that training, medical doctors and like also conduct systematic reviews, and are familiar with MeSH, etc already. And so we strongly contest the argument that this should be written for a non-expert audience or that the non-expert audience is limited to who you may be imagining. And with all due respect, we reiterate that we do not think it is fair that this paper is being judged this way. There are thousands of papers on PLOS ONE that are highly technical and scientific. We’re not sure, therefore, what is motivating this need to appeal to a non-expert audience or why you think that the expert audience is rather limited. Also, even though one of the reviewers pointed out that “library experts already know all this”, we have also shown, and rebutted in our last review, that they only know this anecdotally. Anecdotal knowledge is not scientific knowledge. This should be better evidence than anecdotal. And even if it is known among “library experts”, many of these experts still are not clear in their papers which version of MEDLINE that they are using, as we have discussed in the paper, which is why we make it clear that they should do that in their papers when we address this in our discussion and conclusion. This misunderstanding or lack of knowledge about the problems with MEDLINE also are not necessarily known among other users of MEDLINE, etc., who are experts in some ways with it or do know the basics, but who may not know the problems that we are addressing here.

My suggestion is that you write a paragraph in the introduction to make the following delineations:

Medline bibliographic records

-------- comprise -------

the Medline database

------- which is -------

indexed in different ways

------- by ----------

platforms

------- used for search and access for Medline records -------------

Thank you for your suggestion, but we believe it is clear as it stands.

The point is that access and search over what should be a stable set of Medline records is unreliable and inconsistent depending on anecdotal factors your study has quantified. It might help to think about these issues as if you were a computer programmer who is familiar with the concept of database records, indexing, and information retrieval over structured records. While there are nuances in the way librarians think about and talk about these distinctions, it is often very hard to follow a discussion that conflates these things.

We believe using computer programmer language would only confuse the issue. It’s already difficult to use terms like “database”, which has multiple meanings. We have made every attempt to be clear in our language and we believe we should not be faulted for how people with different field backgrounds might read this.

* You introduce “MESH” on line 64 but do not define or explain it. Drop the reference to LCSH – it is unimportant to your point. Non-expert readers will not make the connection between these two things. Use the space to explain what MESH is and why it matters. Below I also point out places where understanding the MESH concept is critical.

We have dropped the reference to LCSH in this sentence and have added an extra clause that further define MeSH.

* Similarly, in the next two paragraphs you refer to ERIC, DOE, NLM, etc.. Stick to the point you are making without drawing parallels – for your non-expert reader these won’t be meaningful. They just get in the way of understanding your point. Your reader needs to understand the major parts (records, database, index, platform) in order to understand how the system breaks down on a longitudinal basis. By the end of the intro these components should be clear to someone who has never heard of Medline.

We have dropped the references to these parallel systems.

* line 75: don’t say “presumably” – your reader is counting on you to say something does, may, or does not pertain; also, “access points” is librarian talk --- these are index fields

We used the term “access” only and have removed points.

* line 132: don’t say “It seems that” – same point, your reader is counting on you to say something does, may, or does not pertain. Is it true that “many health science informational professionals and librarians are aware of the

133 issues that we report here”? – then just say that.

Revised.

* Much of the material in paragraphs between line 67 and line 81 can be simplified and combined with the clear paragraph that runs from 82 to 89. As suggested above, the parallels are not helpful to your explanation.

Revised and combined.

* You have presented PubMed as a platform in most of the paper but in the introduction you imply that it is an intermediate step between the Medline index and the proprietary platforms. The role of PubMed is muddled. I have thought of two options – and I like my second idea best.

 1) Maybe you can refer to the PubMed WEBSITE as the platform and simply indicate that MESH terms are added to the Medline index before it is distributed to vendors. The fact that the MESH-enhanced index “base” is called PubMed just adds needless confusion about details that are less important (paragraph between lines 82 and 89). This is related to my point on the figures – the nuances are confusing, don’t help your reader understand, and detract from the important findings.

OR

2) Justify using the MESH indexed PubMed as the standard against which other platforms are measured. I think this is a better approach. In this case you need to explain that the MESH-enhanced PubMed index is the foundation used by the other platforms. This makes sense, but you need to get into some version of this:

Medline bibliographic records

-------- comprise -------

the Medline database

------- which is indexed using MESH terms to create

the PubMed database

------- which is further enhanced with additional index fields by the ----------

platforms (including PubMed WEBSITE)

------- that make Medline records accessible for searching -------------

In any event, you are referring to PubMed using two difference constructs and it is confusing. Rather then try to explain the confusion, just use language that clearly separate the two rolls of “PubMed”.

We think our discussion of PubMed is clear as it stands.

* These sentences are very unclear (line 103 – 106): “If searchers used too few databases to conduct literature searches [32–34], then they may miss relevant studies [10,35]. … This is especially important in cases where data from past research is collected, analyzed, and synthesized based on published and/or gray literature,…” – Is this referring to too few platforms? Or are you saying that searchers must use Medline, additional databases, gray literature? These passages are not helping your reader. They go off the track of thinking about how variability in indexes and platforms affects query performance regarding longitudinal consistency. The point of this paragraph is very muddled.

We have divided this into two separate paragraphs and have revised to make it clearer.

* At line 134 you state “We are not aware of what the broader bio-science and medical communities know because few studies have included queries that were designed to test reproducibility be reproducible across systems.” The sentence does not make sense? You’re not aware because of the lack of studies? The revised paragraph is muddled. One issue is that non-experts have little or no information. The other is that prior studies have not been done to measure and confirm the anecdotal knowledge/reports. Both points are important and both are lost in the first part of the paragraph.

We inserted the anecdotal sentence here based on the other reviewer’s comments in the prior review and that you reference above, but this is also anecdotal – that is, whether such library experts are even anecdotally aware of these issues, or if many are, how many or how few are. In many ways, these peer reviews have been very helpful, but they are also causing some of what we are saying to get muddled. So we have removed the anecdotal sentence here because we think it’s caused some of the confusion that you point out here. Besides, we are not sure how well this is anecdotally known, even though the other reviewer claims that it is well known.

* You must explain the concepts of “MESH branch” and “exploded MESH” in language that a database designer, information science person, or computer scientist would use. Understanding your method and results depends on understanding of these concepts. Relate the ideas to the more general notions of index terms, tags, and tree structure.

* The methods section is much easier to follow. Tweaks:

** line 182 “(e.g., all queries including a single keyword search in a journal title field in a case)” is confusing and not needed

Removed.

** line 186 “were designed to search all five MEDLINE platforms” - to search EACH of the MEDLINE platforms

Revised.

** line 189 – that the data WERE collected

Revised.

** lines 193 – 202 – my version: We designed our 29 cases using basic search fields to examine compare search retrieval counts on each of the five MEDLINE platforms, therefore, our queries were designed to be short, logically clear, and to enhance the comparability of the results. For example, Table 1 describes a simple MeSH search, limited by publication date range.

Revised.

** in the lines that follow you report your results as if this is a justification for the method. You can state this more convincingly with something like this:

Our prior findings [cite yourself if this is true – our cite someone else – or simply say “based on our preliminary testing”] indicated that a wide number of records would be retrieved by different platforms, particularly when MESH terms were added to the strategies/queries. On this basis, we created 29 cases then to test a broad range of basic search strategies.

Revised.

** line 235 – “We used the modified z-score to locate deviations around results from the PubMed MEDLINE platform” – this must be justified. Why is PubMed the logical standard? (see above). Explaining this will go a long way toward making the results easier to understand. This is about the indexes. Somewhere be sure you are clearer that indexing and query structure go hand in glove. This helps explain index differences as a source of critical variability in results. Upload timing and errors are easy to understand. Indexing and retrieval differences are not.

Revised.

Conflations of “database”, “index” and “platform”:

* In your abstract, 22-23 “Specific bibliographic databases, like PubMed and MEDLINE….” – but you’ve set this up with MEDLINE as the database – and you present PubMed as one of several platforms. Conflating them in the abstract is very confusing.

We do not think we are not conflating these terms. In the first sentence, this is clear with “… of the MEDLINE database hosted on multiple platforms”.

* line 31 “Bibliographic databases are used to identify and collect research data… ” – here you are talking about platforms, not databases

“Bibliographic databases” is a legitimate term for this. Here we are not referring to any platform but to bibliographic databases generally.

* line 33 “These systems, in particular, PubMed and MEDLINE, are used to inform clinical….” – just say “these systems” to avoid confusion about the differences between PM and ML. Stick with the simple, clear and correct but incomplete definition of Medline as a core underlying database of bibliographic records and PubMed as a platform that has its own index of the Medline records

Removed “These systems”, but PubMed is more than MEDLINE and includes records not indexed in MEDLINE.

* line 36 “…the development of bibliographic databases or…” – the development of bibliographic platforms

This is not necessary to change. Platforms are much more than the bibliographic databases. Here we are referring to databases.

* line 58 “…bibliographic databases, such as those available on…” – you are talking about indexes here

We are talking about databases that exist on the platforms. We lightly revised.

* lines 62-63 “In some specialized databases, these records may be supported” – you are talking about indexes here

No, a database creates an index. We’re talking about databases. Our language is correct.

* line 90 “This differentiation among database platforms has” – this is differentiation between platforms

Revised.

* line 97 “the same MEDLINE data file” – same MEDLINE database

Revised.

* line 101 “Although the choice of database systems impacts potential source” – this is the choice of platforms

Revised.

* line 102 “it is not known how searching the sdame database (e.g., MEDLINE) on different platforms” – simplify this === “it is not known how searching .. the MEDLINE database… on different platforms … ” – that’s what you are talking about

Revised.

* line 117 “principles are applied by database vendors in indexing” -- these are platform vendors – or just platforms

Revised.

* line 120 “problematic in bibliographic databases purchased by libraries” – bibliographic platforms – ore more generally “bibliographic resources” – reserve “database” to refer to Medline only

Revised.

* line 125 “license these records to database vendors to host on their own platforms” – to vendors – you don’t need to call them database vendors

Revised.

Finally, your results section is still extremely unclear and profusion of charts is distracting and confusing. All this seriously undermines your point. The notion of macro and micro are also not useful. Rather then giving these two perspectives unnecessary and confusing names, just write what what you are talking about – what you did. The idea of analysis “scale” is also not useful. In order to understand what you are reporting I need to revert to looking at your raw data. In short, major changes are still needed to make the results meaningful for non-expert readers. I believe only a determined expert will be motivated to decipher it.

We disagree that this is confusing and that the terms we’re using are not useful. We already know that our results have been looked at based on the comments via social media about our preprint (https://www.biorxiv.org/content/10.1101/2020.05.22.110403v1) and that our points have been understood. Figure 3 is an important contrast to Figures 4, 5, and 6, and the terms “micro” and “macro” have no special meaning beyond their everyday definitions. However, based on the other reviewer’s comments, we have clarified the language and research involving the macro and micro views.

In general, Figure 3 is meaningless. You can show one example if you wish, but the text should convey that whatever it is you are measuring in those charts is a constant month over month. The results displayed in Figures 4, 5 and 6 are meaningful, but the meaning is buried in the complexity of the text and in all the charts. In my opinion, your results section should focus on making very clear the meaning and implications of data displayed in figs. 4, 5, 6, but you need to select only one chart as an example for each type of anomaly you have found. You can make all the other charts available online with the data. The details in figures 7 and 8 are also obscure your point. You can report that information much more effectively with one example figure and a summary table on your stats.

We strongly disagree that these figures are meaningless or that they obscure our point. If you feel you need to recommend rejecting this article for these reasons, I guess go ahead. We’ve already defended these plots and we think they are important visualizations. 

Many of the details of your results section are dense and hard to understand, particularly without definitions of things like MESH and “explode”. If there are important esoteric nuances to the specific cases, that detail may be better suited for publication in a second paper written specifically for experts.

Again, we think this criticism is unfair and does not impact the soundness of our research. We are not writing for the general public, just as most articles on PLOS are not for the general public. We appreciate your recommendations above and previously. You have helped us clarify the language and the terms, which are confusing for a number of reasons beyond this research. But we can only go so far in making this paper accessible to the “non-expert”. 

With a greatly simplified presentation of your results readers will be able to follow. This will make your final chart (Fig 9) much more meaningful. This is a nice, clear and current demonstration of the effect of the issues you’ve characterized.

Reviewer #3: This is my second review of your paper. You and your team have made progress in some areas of the paper; some parts of the paper are coming together, but other parts are still unclear.

I've made extensive editorial comments (N=34 in total) focusing on using simpler, more consistent language in the manuscript. Some sections in the paper are still difficult to understand; I've done my best to try to assist in clarifying your points but not certain I've achieved much. Best of luck with your research.

My comments are:

1) I like the new title for the article, "MEDLINE search retrieval issues: A longitudinal query analysis across select vendor platforms" though I would take out the word 'select' and replace it with the word, "five".

Revised.

2) Abstract: Your first sentence is too long and somewhat confusing. Can you consider a change to the second sentence at line 14 as, "We devised twenty-nine search queries, or cases, comprised of five semantically equivalent queries per case to search against five MEDLINE platforms." Lines 17-23 are important and I'd recommend breaking up the section by saying, "We found that search results varied by MEDLINE platform within sets and across time. Reasons for variations were due to trends in scholarly publishing such as publishing individual papers online first versus complete issues in print format. Some other reasons were metadata differences in bibliographic records; differences in levels of specificity of search fields across platforms and large fluctuations in monthly search results based on the same query. Database integrity and currency issues were observed as each platform updated its MEDLINE files throughout the year."

Revised.

3) Line 35 should say "...in the health professions and to construct a knowledge base for clinical decision support systems".

Revised.

4) Line 49 remove "should".

Revised.

5) Line 53 sounds a bit odd with "relies"; I would use rely for better agreement.

Revised.

6) Lines 54-58 could be shortened as "The Preferred Reporting Items for Systematic Reviews and Meta-Analyses (PRISMA) Guidelines and the Cochrane Handbook for Systematic Reviews of Interventions provide examples for scholars in their reporting of methods and organizing of reviews".

7) At line 60, it's not that they rely on structured records but that they are designed that way for descriptive purposes and to create reliable post-hoc searchable citations/ records.

Revised.

8) This sentence is unnecessarily awkward at line 61, "These bibliographic records contain fields we take as meaningful for providing discovery and access, such as author name fields, document title fields, publication title fields, and date of publication fields." Perhaps take out "we take as meaningful'.

Revised.

9) At line 65, I would not use this phrase, "are based on standard knowledge classification efforts". I would not refer to subject analysis and/or article indexing as knowledge classification for obvious reasons.

Revised.

10) Line 75, I don't really understand why you use the phrase "Commercial information service companies" when you can say simply "Commercial vendors such as" or simply "vendors". I wouldn't end the sentence at line 78 with a preposition. I would say "database providers, such as the NLM or DOE, etc" and remove the word 'original' which is not accurate in any case.

Revised based on your comment and the other reviewer’s.

11) Line 78, isn't it really about database "features" and not "value"? What do you mean by value? Not sure I agree that it's about value and not about searchable features.

Just that adding features that are different than what can be found on PubMed is a selling point for them. But revised.

12) Commercial information service provider at line 86, say "vendor". You don't need to keep changing the phrase which will only confuse your reader. (You also continue this practice throughout the manuscript).

Revised throughout.

13) Line 90 why say "differentiation"? when you can say "Differences in features across platforms"? And I'm not sure that accounting for those differences is why we report the specific platforms, interfaces and / or vendors. It's to help reproduce the searches as reported by using the correct platform.

Revised.

14) Line 97 I would remove "presumably". It's a kind of cynical word to use in this context.

Revised.

15) Line 101: this sentence seems to relate more to the previous sentence than what comes after it.

Revised this section.

16) Line 114: "database systems" should be vendor or "vendor systems" or "database vendors". If clarity is needed, you can say "Vendor systems / or vendors such as OVID or EBSCOHost".

Revised.

17) Line 121: I'd remove "also". There are too many examples of unneeded uses of this word in your paper; many are not needed.

Revised.

18) Line 132: Remove "It seems that" and the sentence needs a citation. I would argue that varying results by vendor platform have been seen in other papers and you could cite one or two. Cite your previous paper from JMLA for example.

Revised based on this and other other reviewer’s comments.

19) Line 144: Change this sentence to something along the lines of "This paper builds on that work and examines differences over time by evaluating longitudinal data, which is a critical factor in replicating bibliographic database search results. "

Revised.

20) Line 162: This first sentence is terribly confusing to me. I would say, "We designed 29 cases, or sets, of searches which comprised five (5) semantically similar or equivalent queries to perform searches in five MEDLINE platforms for a total of 145 searches".

We have taken out the term sets because another reviewer didn’t like that term and suggested using cases. Otherwise revised.

21) Lines 180-188: can you say here why you picked dementia? Because it's one word? Because it is short and easy? Some rationale here is really needed even if you feel you are repeating information. It will help your reader understand why you picked it and grouped it with neoplasia.

Revised. Chosen just based on simplicity and for representation. The term was not grouped with neoplasms. Those were separate searches.

22) Line 247: Can you set up this section for clarity? For example, "The data gathered in the majority of our searches (20 cases of five searches each for a total of 100 queries) did not include limits on publication dates. However, we did include date delimited searches for nine cases of five searches for a total of 45 queries. We used publication date limits from 1950 to 2015 for these search queries. We found that...etc."

Revised.

23) Line 259: This sentence is awkward. Can you say simply, "Second, based on a detailed examination of the same cases...". Remove the use of the word also (2x) from lines 263 and 268.

Revised.

24) Line 272: This whole section on macro/micro is more confusing than it needs to be. I would consider using more simple language. For example "We examined the data of the cases, and present variations we identified based on two categories which we will use in the following section: an overview (macro) of the data and a more granular (or, micro) view of the variations arising in the data".

Thank you. Revised.

25) Line 323: I would relabel this section as "Online first and print publications impact bibliographic control". Also line 327: I would avoid the phrase "non exploding". Can you say "...not exploded, but differed because case #04 is restricted by publication date". Please use simpler sentence construction for the benefit of communicating your results. Throughout this section, I would look critically at using the phrase "non exploding" as this is not the way this is referred to in describing the feature. Say "not exploded".

Revised throughout.

26) Line 339: I would not say "indicates"; perhaps "suggests" would make more sense.

Revised.

27) Line 348: Change along these lines, "The record indicates it was added to PubMed in 2015 but not formally entered into MEDLINE until 2019".

Revised.

28) Lines 361-378 should be edited for brevity. It's also very acronym heavy.

Revised.

29) Line 379: what do you mean by the title of this section "Reproducible with specific field searches"? Do you mean "Reproducibility improved with search specificity (or field searching)"

Yes. Revised.

30) Lines 394-397: the label for this section is a bit unclear as is the first sentence of the section. Perhaps it's the use of the word "outliers". What does this mean? Just different from the rest? An error? An incorrectly inflated number?

Revised.

31) Lines 400-522: this section is quite difficult to read. Without going into detail, I'd recommend a close edit. For example, at Line 488-490, try something simpler such as "We could not identify the reasons for the significant variance in search results over time. As such, searchers may be left wondering why they are seeing variances in search results which may further undermine their trust in using the different platforms".

Revised.

32) In your Limitations section, you need an introductory sentence. "Our paper has a number of limitations based on X, Y and Z." As it's written in the first sentence, I don't understand what you mean by saying that using PubMed / Medline is a limitation (due to different coverage and content or...?).

We removed the first paragraph. 

33) Line 523: why is documenting variances in systematic searches across platforms a useful area of research? Could you explain this to the reader? It might be important - but your explanation doesn't really clarify. Perhaps you could say that documenting variances on platforms is NOT as important as trying to identify which vendor platform is the most reliable over time, and why. That's the issue we as searchers want to know about and why we are reading your paper. Perhaps a ranked list would be a good research project. It shouldn't be up to health librarians or expert searchers to document these variances. They take up our time! to do searches. I think the semantically selected search queries are a limitation also but you don't really know because you haven't tested them. You might say this may have affected generalizability.

Revised and explained. We think we explained the importance in the beginning of the paper but we have attempted to tie up that loose end. Regarding the semantic issue, we’re not sure what you mean. But also, this is not something that will be generalizable, as we discussed in the previous review. The systems need to be standardized.

34) Conclusions in your paper: Not sure why you introduce Google Scholar in your lead sentence of your conclusion. I'd reconsider that first sentence very closely. Also, what's the most important concluding remark about your research after all the work you put into it? Perhaps it's that "No single MEDLINE platform emerged as the most reliable interface to search for citations for this project". The data suggests that there were greater variances in search results over time for simpler, shorter search queries and that the more sophisticated search queries performed more reliably over the five vendor platforms. Whatever it is, can you be explicit? Best of luck.

/dg

We say that this study indicates that there is no one MEDLINE. We’re not sure the critique about the analogy to Google Scholar.

Revised.

27) Line 348: Change along these lines, "The record indicates it was added to PubMed in 2015 but not formally entered into MEDLINE until 2019".

Revised.

28) Lines 361-378 should be edited for brevity. It's also very acronym heavy.

Revised.

29) Line 379: what do you mean by the title of this section "Reproducible with specific field searches"? Do you mean "Reproducibility improved with search specificity (or field searching)"

Yes. Revised.

30) Lines 394-397: the label for this section is a bit unclear as is the first sentence of the section. Perhaps it's the use of the word "outliers". What does this mean? Just different from the rest? An error? An incorrectly inflated number?

Revised.

31) Lines 400-522: this section is quite difficult to read. Without going into detail, I'd recommend a close edit. For example, at Line 488-490, try something simpler such as "We could not identify the reasons for the significant variance in search results over time. As such, searchers may be left wondering why they are seeing variances in search results which may further undermine their trust in using the different platforms".

Revised.

32) In your Limitations section, you need an introductory sentence. "Our paper has a number of limitations based on X, Y and Z." As it's written in the first sentence, I don't understand what you mean by saying that using PubMed / Medline is a limitation (due to different coverage and content or...?).

We removed the first paragraph. 

33) Line 523: why is documenting variances in systematic searches across platforms a useful area of research? Could you explain this to the reader? It might be important - but your explanation doesn't really clarify. Perhaps you could say that documenting variances on platforms is NOT as important as trying to identify which vendor platform is the most reliable over time, and why. That's the issue we as searchers want to know about and why we are reading your paper. Perhaps a ranked list would be a good research project. It shouldn't be up to health librarians or expert searchers to document these variances. They take up our time! to do searches. I think the semantically selected search queries are a limitation also but you don't really know because you haven't tested them. You might say this may have affected generalizability.

Revised and explained. We think we explained the importance in the beginning of the paper but we have attempted to tie up that loose end. Regarding the semantic issue, we’re not sure what you mean. But also, this is not something that will be generalizable, as we discussed in the previous review. The systems need to be standardized.

34) Conclusions in your paper: Not sure why you introduce Google Scholar in your lead sentence of your conclusion. I'd reconsider that first sentence very closely. Also, what's the most important concluding remark about your research after all the work you put into it? Perhaps it's that "No single MEDLINE platform emerged as the most reliable interface to search for citations for this project". The data suggests that there were greater variances in search results over time for simpler, shorter search queries and that the more sophisticated search queries performed more reliably over the five vendor platforms. Whatever it is, can you be explicit? Best of luck.

/dg

We say that this study indicates that there is no one MEDLINE. We’re not sure the critique about the analogy to Google Scholar.

---

## [Decision Letter · Decision Letter 3]

23 Feb 2021

PONE-D-20-15022R3

MEDLINE search retrieval issues: A longitudinal query analysis of five vendor platforms

PLOS ONE

Dear Dr. Burns,

Thank you for submitting your manuscript to PLOS ONE. After careful consideration, we feel that it has merit but does not fully meet PLOS ONE’s publication criteria as it currently stands. Therefore, we invite you to submit a revised version of the manuscript that addresses the points raised during the review process.

As you can see in the review report at the end of this email, both reviewers think that their comments have been adequately addressed. However, one reviewer has suggested minor language improvements. So, it is recommended that the manuscript is thoroughly proofread for typos and any language errors. 

We look forward to receiving your revised manuscript.

Kind regards,

Muhammad A. Z. Mughal, PhD

Academic Editor

PLOS ONE

Journal Requirements:

Reviewers' comments:

Reviewer's Responses to Questions

**Comments to the Author**

1. If the authors have adequately addressed your comments raised in a previous round of review and you feel that this manuscript is now acceptable for publication, you may indicate that here to bypass the “Comments to the Author” section, enter your conflict of interest statement in the “Confidential to Editor” section, and submit your "Accept" recommendation.

Reviewer #1: All comments have been addressed

Reviewer #3: All comments have been addressed

2. Is the manuscript technically sound, and do the data support the conclusions?

Reviewer #1: Yes

Reviewer #3: Partly

3. Has the statistical analysis been performed appropriately and rigorously? 

Reviewer #1: Yes

Reviewer #3: Yes

4. Have the authors made all data underlying the findings in their manuscript fully available?

Reviewer #1: Yes

Reviewer #3: Yes

5. Is the manuscript presented in an intelligible fashion and written in standard English?

Reviewer #1: Yes

Reviewer #3: Yes

6. Review Comments to the Author

Reviewer #1: You have addressed my concerns. Thanks for all your hard work improving the paper. I believe it makes a valuable contribution.

Reviewer #3: Dear Authors,

This is now my third review of your manuscript, now entitled "MEDLINE search retrieval issues: A longitudinal query analysis of five vendor platforms". I agree your paper is improved but wonder if it would benefit even further from another vigorous editing. Show it to a copyeditor because it's important research. (It's possible the peer review process has forced you to make changes to your paper that conflict in some way with what you are communicating, which has happened to me before.)

In any case, here are some of my suggestions for your consideration:

1. Abstract, first sentence: Could you not say, "This study compares results of a longitudinal search query analysis of

MEDLINE hosted on five (5) platforms: PubMed, EBSCOHost, Ovid, ProQuest, and Web of Science." In reading the sentence as it is currently, it sounds as though you are covering more than five platforms. ("....that include").

2. Abstract, fifth sentence: could you put in past tense? "We found that search results varied considerably depending on MEDLINE platform."

3. Abstract, later sentence: Could you not say "biomedical" instead of "bioscience"? Otherwise you have to define what you mean by this unusual word. Do you mean biomedical sciences? Instead, how about: "....Specific biomedical bibliographic databases are used to inform clinical decision-making, create systematic review searches....etc."

4. Abstract, last sentence: could you not simplify as "They serve as essential information retrieval and discovery tools to help identify and collect research data, and are used in a broad range of fields as the basis of research design. This study should help clinicians, researcher, librarians, informationists, and others understand how these five platforms differ and inform future work in their standardization." (I have not heard "informationalists" before and didn't catch this in the earlier version.)

5. Introduction, first paragraph, this is your all-important entry point for your reader. Could you consider something along the lines of, "Bibliographic databases are used to identify and collect research papers and function as a critical part of scientific investigation. Studies that employ bibliographic databases include research on bibliometrics/scientometrics, information literacy, systematic reviews, and meta-analyses, to name a few. In particular, PubMed and MEDLINE are used to inform clinical decision-making by health professionals and in building knowledge bases for clinical decision support systems". (I see you cite the Hines (2006) paper but remember that study is about genomic databases not bibliographic databases; I would not extrapolate and say that bibliographic databases are scientific instruments. Sorry I didn't capture that beforehand.)

6. Line 41, remove the extra "to"s...."Researchers, librarians, information scientists, and others rely on bibliographic databases to conduct research, instruct future information and other professionals on how to conduct literature searches, and assist those with information needs to locate and access literature"

7. Line 44, this sentence will confuse your reader and should be simplified. Could you consider, "Furthermore, these databases have standard rules to describe research papers, and are structured using controlled vocabularies, in order to make searching for information more precise or comprehensive".

8. Line 47, could you say "Fine control over bibliographic searching and documentation of search strategies, which are reported in systematic reviews, allow for the replication and reproduction of searches."

9. Line 61, could you break into two sentences such as, "In certain specialized databases, these records are often supported by a set of thesaurus terms, or a controlled vocabulary, such as the Medical Subject Headings (MeSH) in the MEDLINE database. Their goal is to describe the subject matter of works or records that are indexed in the bibliographic database to assist in consistent information retrieval."

10. Line 67, amend slightly as "The MeSH thesaurus is freely available on the U.S. National Library of Medicine's (NLM) PubMed website, and is used to search MEDLINE on other platforms such as EBSCOhost, Ovid, ProQuest, and Web of Science."

11. Line 70, I would say MEDLINE here not PubMed. Vendors buy MEDLINE not PubMed, "These commercial vendors provide access to the bibliographic data from MEDLINE and the corresponding MeSH thesaurus and add features on their respective platforms beyond what NLM has already provided."

12. Line 72, repetition of "their" can be removed and amended, "The added features are based on a vendor's unique user interface, search features, ability to link to library collections via proxies, related additional database content, or searching multiple databases on a specific platform in a single search session."

13. Line 76, paragraph amended along the lines of, "However, these added features may create some differences in searching and in the search results [26,27]. For example, MEDLINE can be searched using PubMed which is defined by the nearly 6000 publications it indexes, structure of its bibliographic records, use of MeSH in those records, and its freely-available search interface on the web. When vendors provide access to MEDLINE, they start with the MEDLINE system but creating a customized subscription-based version that includes a new interface, slightly different search fields, search operators, and other features."

Note to authors: I regret that I have run out of time. Best of luck with your manuscript.

7. PLOS authors have the option to publish the peer review history of their article (what does this mean?). If published, this will include your full peer review and any attached files.

Reviewer #1: No

Reviewer #3: No

---

## [Author Response · Author response to Decision Letter 3]

1 Mar 2021

Dear Editor,

We have accepted all the suggestions from Reviewer #3 below, and per their additional advice, have thoroughly edited the manuscript. We would like to extend our gratitude to you and the reviewers for their close inspection of the manuscript and for their patience with this review process.

Sincerely,

C. Sean Burns

Reviewer #3: Dear Authors,

This is now my third review of your manuscript, now entitled "MEDLINE search retrieval issues: A longitudinal query analysis of five vendor platforms". I agree your paper is improved but wonder if it would benefit even further from another vigorous editing. Show it to a copyeditor because it's important research. (It's possible the peer review process has forced you to make changes to your paper that conflict in some way with what you are communicating, which has happened to me before.)

In any case, here are some of my suggestions for your consideration:

1. Abstract, first sentence: Could you not say, "This study compares results of a longitudinal search query analysis of

MEDLINE hosted on five (5) platforms: PubMed, EBSCOHost, Ovid, ProQuest, and Web of Science." In reading the sentence as it is currently, it sounds as though you are covering more than five platforms. ("....that include").

2. Abstract, fifth sentence: could you put in past tense? "We found that search results varied considerably depending on MEDLINE platform."

3. Abstract, later sentence: Could you not say "biomedical" instead of "bioscience"? Otherwise you have to define what you mean by this unusual word. Do you mean biomedical sciences? Instead, how about: "....Specific biomedical bibliographic databases are used to inform clinical decision-making, create systematic review searches....etc."

4. Abstract, last sentence: could you not simplify as "They serve as essential information retrieval and discovery tools to help identify and collect research data, and are used in a broad range of fields as the basis of research design. This study should help clinicians, researcher, librarians, informationists, and others understand how these five platforms differ and inform future work in their standardization." (I have not heard "informationalists" before and didn't catch this in the earlier version.)

5. Introduction, first paragraph, this is your all-important entry point for your reader. Could you consider something along the lines of, "Bibliographic databases are used to identify and collect research papers and function as a critical part of scientific investigation. Studies that employ bibliographic databases include research on bibliometrics/scientometrics, information literacy, systematic reviews, and meta-analyses, to name a few. In particular, PubMed and MEDLINE are used to inform clinical decision-making by health professionals and in building knowledge bases for clinical decision support systems". (I see you cite the Hines (2006) paper but remember that study is about genomic databases not bibliographic databases; I would not extrapolate and say that bibliographic databases are scientific instruments. Sorry I didn't capture that beforehand.)

6. Line 41, remove the extra "to"s...."Researchers, librarians, information scientists, and others rely on bibliographic databases to conduct research, instruct future information and other professionals on how to conduct literature searches, and assist those with information needs to locate and access literature"

7. Line 44, this sentence will confuse your reader and should be simplified. Could you consider, "Furthermore, these databases have standard rules to describe research papers, and are structured using controlled vocabularies, in order to make searching for information more precise or comprehensive".

8. Line 47, could you say "Fine control over bibliographic searching and documentation of search strategies, which are reported in systematic reviews, allow for the replication and reproduction of searches."

9. Line 61, could you break into two sentences such as, "In certain specialized databases, these records are often supported by a set of thesaurus terms, or a controlled vocabulary, such as the Medical Subject Headings (MeSH) in the MEDLINE database. Their goal is to describe the subject matter of works or records that are indexed in the bibliographic database to assist in consistent information retrieval."

10. Line 67, amend slightly as "The MeSH thesaurus is freely available on the U.S. National Library of Medicine's (NLM) PubMed website, and is used to search MEDLINE on other platforms such as EBSCOhost, Ovid, ProQuest, and Web of Science."

11. Line 70, I would say MEDLINE here not PubMed. Vendors buy MEDLINE not PubMed, "These commercial vendors provide access to the bibliographic data from MEDLINE and the corresponding MeSH thesaurus and add features on their respective platforms beyond what NLM has already provided."

12. Line 72, repetition of "their" can be removed and amended, "The added features are based on a vendor's unique user interface, search features, ability to link to library collections via proxies, related additional database content, or searching multiple databases on a specific platform in a single search session."

13. Line 76, paragraph amended along the lines of, "However, these added features may create some differences in searching and in the search results [26,27]. For example, MEDLINE can be searched using PubMed which is defined by the nearly 6000 publications it indexes, structure of its bibliographic records, use of MeSH in those records, and its freely-available search interface on the web. When vendors provide access to MEDLINE, they start with the MEDLINE system but creating a customized subscription-based version that includes a new interface, slightly different search fields, search operators, and other features."

---

## [Editor Report · Decision Letter 4]

29 Mar 2021

MEDLINE search retrieval issues: A longitudinal query analysis of five vendor platforms

PONE-D-20-15022R4

Dear Dr. Burns,

We’re pleased to inform you that your manuscript has been judged scientifically suitable for publication and will be formally accepted for publication once it meets all outstanding technical requirements.

Kind regards,

Muhammad A. Z. Mughal, PhD

Academic Editor

PLOS ONE
---

## [Editor Report · Acceptance letter]

21 Apr 2021

PONE-D-20-15022R4 

MEDLINE search retrieval issues: A longitudinal query analysis of five vendor platforms 

Dear Dr. Burns:

I'm pleased to inform you that your manuscript has been deemed suitable for publication in PLOS ONE. Congratulations! Your manuscript is now with our production department. 

Kind regards, 

on behalf of

Dr. Muhammad A. Z. Mughal 

Academic Editor

PLOS ONE